

# Ground-state phase diagram
# of quantum link electrodynamics in $(2+1)$-d

Tomohiro Hashizume[1], Jad C. Halimeh[2], Philipp Hauke[2] and Debasish Banerjee[3]

**1** Department of Physics and SUPA, University of Strathclyde,
Glasgow G4 0NG, United Kingdom
**2** INO-CNR BEC Center and Department of Physics, University of Trento,
Via Sommarive 14, I-38123 Trento, Italy
**3** Saha Institute of Nuclear Physics, HBNI, 1/AF Bidhannagar,
Kolkata 700064, India

## Abstract

The exploration of phase diagrams of strongly interacting gauge theories coupled to matter in lower dimensions promises the identification of exotic phases and possible new universality classes, and it facilitates a better understanding of salient phenomena in Nature, such as confinement or high-temperature superconductivity. The emerging new techniques of quantum synthetic matter experiments as well as efficient classical computational methods with matrix product states have been extremely successful in one spatial dimension, and are now motivating such studies in two spatial dimensions. In this work, we consider a U(1) quantum link lattice gauge theory where the gauge fields, represented by spin-$\frac{1}{2}$ operators are coupled to a single flavor of staggered fermions. Using matrix product states on infinite cylinders with increasing diameter, we conjecture its phase diagram in $(2+1)$-d. This model allows us to smoothly tune between the U(1) quantum link and the quantum dimer models by adjusting the strength of the fermion mass term, enabling us to connect to the well-studied phases of those models. Our study reveals a rich phase diagram with exotic phases and interesting phase transitions to a potential liquid-like phase. It thus furthers the collection of gauge theory models that may guide future quantum-simulation experiments.

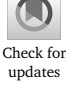
# 1 Introduction

Quantum field theories involving matter fields coupled with gauge fields provide a theoretical framework to explore and understand a host of phases of matter occurring in nature, from the very low to the very high energy scales. Quantum electrodynamics (QED) [1] is probably the most well-known interacting gauge theory in $(3 + 1)$−d, which describes the interaction of photons (via a U(1) gauge field) with electrons (or matter in general). This theory is as necessary to understand scattering of sunlight in the atmosphere as it is to operate a Tokamak [2]. Similarly, Fermi's theory of beta-decay [3] evolved into an SU(2) gauge theory of electroweak interactions [4], which explained diverse phenomena from the decay of the nucleus to the breaking of charge conjugation and parity. The SU(3) gauge theory of quantum chromodynamics (QCD) [5] explains the wide phenomenology of strong interactions, provides a framework to compute the mass of the proton and other hadrons from first principles, and predicts the existence of a quark gluon plasma at high temperatures and neutron stars at high densities.

While these are considered to be traditional high-energy physics examples of naturally occurring gauge theories, there are equally varied applications of gauge theories in condensed-matter physics. In fact, the notion of emergent gauge fields is extensively used to explain a variety of physical phenomena. Deconfined quantum criticality [6] describes phase transitions that can occur outside the Landau paradigm of phase transitions. They are characterized by order parameters that become deconfined only at the critical point, necessitating a gauge-theory description. Topological insulators can be described by an effective theory of electrodynamics with a non-zero theta angle (equal to $\pi$) [7]. Ferromagnetic superconductivity [8] and the fractional quantum Hall effect [9] are other well-known examples.

The phases of such interacting gauge theories can thus give rise to extremely varied phenomena, not only important to understand the functioning of Nature, but also useful to further our technological progress. One of the biggest roadblocks in our efforts to exploit this richness is the lack of universally applicable computational methods. Due to the weak-coupling nature of quantum electrodynamics, a lot of associated phenomena can be studied analytically, or through the clever use of perturbation theory and other weak-coupling methods. Most of

the gauge theories interacting with matter, however, are strongly interacting, and there is no universal method to study them in a controlled way.

Nevertheless, the past decades have witnessed an exciting development of theoretical tools that have greatly increased our power of addressing such strongly interacting systems. Quantum Monte Carlo methods have been significantly advanced to address a host of systems also beyond quantum chromodynamics (QCD) [10]. For example, novel cluster and worm algorithms are available to simulate a class of bosonic and spin systems [11]. The meron [12], fermion bag [13], and determinantal algorithms [14] for fermions have made exotic phases and universality classes accessible to us. Simultaneously, new classes of gauge theories, known as quantum link models, which realize continuous gauge invariance with finite dimensional Hilbert spaces [15], have widely extended the conventional reach of Wilson's lattice gauge theories [16]. Novel phases have been uncovered in such link models [17, 18] using newly developed quantum Monte Carlo algorithms [19].

A complementary theoretical tool, the concept of density matrix renormalization group [20–22] and its application to matrix product states [23], have proved to be a powerful development that can handle a host of cases, especially in lower dimensions, and even in infinite volumes [24, 25]. They can even be used when existing Monte Carlo methods fail due to a sign problem, and can also be applied to study the dynamics of quantum systems in real-time [26, 27]. Tensor Network methods for such strongly interacting lattice gauge theories are also under extensive investigation [28, 29].

Beyond the realm of classical computing, a complete paradigm shift has appeared in the past two decades through the development of quantum computing methods, both analog and digital. This approach takes Feynman's suggestion [30] of using quantum degrees of freedom to model and simulate quantum systems on quantum tabletop experiments [31, 32], which now are also being deployed on an engineering scale by Google [33–36], IBM [37], Rigetti [38], IonQ [39], Alpine Quantum Computing [40], Pasqal [41], and others. Analog and digital simulation experiments are maturing rapidly and are routinely used to realize a host of Hamiltonians of interest to theorists on tabletop experiments [42–47]. These new computing abilities generate a huge incentive to explore the properties of a variety of models involving gauge fields strongly interacting with fermions, which are out of reach for current classical computations. Not only does this possibility offer the prospect of exploring phases occurring in natural substances, but it also provides a tunable environment which allows us to adjust energy scales to simplify the physics and isolate the features of interest. Simultaneously, the plethora of models accessible in quantum devices can become useful for experimentally demonstrating certain concepts or exotic phases that were theoretically imagined first, such as deconfined quantum criticality.

In this article, we explore such a strongly interacting fermionic theory interacting with Abelian gauge fields in two spatial dimensions. The gauge fields are realized as spin-$\frac{1}{2}$ quantum links, and therefore the entire model has a local finite-dimensional Hilbert space, which lends itself to implementations in quantum simulator experiments. In Sec. 2, we present the model as well as the local and the global symmetries under which it is invariant. In Sec. 3, we discuss the numerical method used to study the model for a range of couplings, namely the infinite density matrix renormalization group technique. This method enables us to take one of the two spatial dimensions to infinity. In Sec. 4, we map out the phase diagram of the model as a function of the bare parameters by introducing order parameters sensitive to various symmetry breakings. Based on the behavior of the order parameter, as well as the correlation function of various fermionic and gauge field operators, we offer a portrait of the phase diagram. Interestingly, we find that our model interpolates between two well-studied pure gauge theories, the U(1) pure quantum link model (QLM) [17] and the quantum dimer model (QDM) on the square lattice [48–50]. Further, we uncover a region in the phase di-

agram that is indicative of a disordered liquid-like phase, also suggested in recent studies of a similar model [51]. This region is separated from the pure gauge QLM-like and QDM-like phases by lines with large correlation lengths, suggestive of two successive quantum phase transitions in the thermodynamic limit. Finally in Sec. 5, we discuss our results and offer an outlook for future studies, both theoretically and experimentally in quantum simulators. Our work is further supplemented with Appendix A for a discussion of the winding number, Appendix B for results on convergence with bond dimension, Appendix C for an analysis of system-size dependence, and Appendix D for a comparison with exact diagonalization results.

Our studies thus reveal a rich ground-state phase diagram of this $(2 + 1)$-dimensional Abelian lattice gauge theory, which we hope will stimulate future numerical and laboratory investigations.

# 2 Model and local constraints

In this section, we introduce the Hamiltonian and gauge-symmetry generators of the lattice gauge theory considered, a quantum link model [51–54] in (2+1)-dimensions with U(1) gauge fields coupled to a single fermionic flavor of matter. We also discuss the global symmetries of the model, which guide the possible phase and symmetry-breaking patterns we can expect.

As mentioned in the Introduction, we use the quantum link formulation of lattice gauge theories for our investigations. It is useful to keep in mind that while it is possible to realize the physics of the Wilson formulation with quantum links for example through dimensional reduction or through scaling $S \to \infty$ [55–57], we limit ourselves to the case of $S = 1/2$, where the physics is the most distinct from the Wilson theory. As we show later, this implies that the electric field energy no longer enters the Hamiltonian as a relevant coupling, but it still plays a crucial role in determining the physics through the Gauss law. Moreover, in our studies, we do not look for the traditional continuum limit (by appropriately scaling the lattice spacing and the bare coupling), but seek to identify the stable thermodynamic phases of the lattice theory in different parameter regimes, which can serve as effective field theory descriptions for the physics of superconductors and spin-ice materials. We emphasize that such studies are also useful to uncover second order phase transitions between different phases, where a continuum limit can be taken, different from the traditional one. In particular, some of such continuum limits can give rise to non-relativisitic theories. Not only does such studies bring into focus the wide class of different physics scenarios that lattice gauge theories host, it also provides a clear motivation for realizing these scenarios in quantum simulator experiments.

## 2.1 Hamiltonian and Gauss's law

The model is defined on a square lattice with fermionic matter degrees of freedom located at the sites, as shown in Fig. 1a. The interactions between fermions are mediated by the gauge fields residing on the bonds connecting the sites. The matter degrees of freedom are single-component spinless fermions, created and annihilated respectively by the operators $\hat{\psi}_{\mathbf{j}}^{\dagger}$ and $\hat{\psi}_{\mathbf{j}}$ on site $\mathbf{j} = (j_x, j_y)$. They satisfy the anti-commutation relations $\{\hat{\psi}_{\mathbf{i}}^{\dagger}, \hat{\psi}_{\mathbf{j}}\} = \delta_{\mathbf{i},\mathbf{j}}$ and $\{\hat{\psi}_{\mathbf{i}}, \hat{\psi}_{\mathbf{j}}\} = \{\hat{\psi}_{\mathbf{i}}^{\dagger}, \hat{\psi}_{\mathbf{j}}^{\dagger}\} = 0$.

Denoting the unit vectors in the $x$ and $y$ spatial directions as $\mathbf{e}_x$ and $\mathbf{e}_y$, the gauge-field operators $\hat{U}_{\mathbf{j},\mathbf{e}_{\mu}}$ and $\hat{U}_{\mathbf{j},\mathbf{e}_{\mu}}^{\dagger}$ reside on the bond joining the sites $\mathbf{j}$ and $\mathbf{j} + \mathbf{e}_{\mu}$. The canonically conjugate momenta are the electric flux operators $\hat{E}_{\mathbf{j},\mathbf{e}_{\mu}}$ and satisfy the following commutation

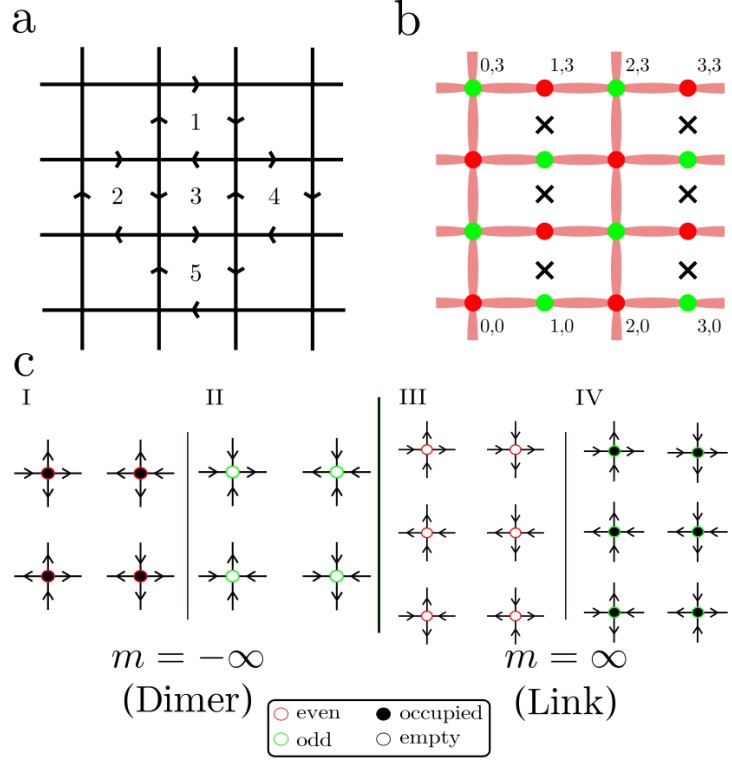

Figure 1: a. Layout of the square lattice with sites and links. The link configuration depicts some flippable plaquettes (as explained in the text), denoted as 1,2,4,5. Flipping any of the plaquettes destroys the flippability of those plaquettes that share a link. b. The phase factors of the links of the square lattice, $s_{\mathbf{j},\mathbf{e_x}}$ are 1 for all the horizontal links, and $s_{\mathbf{j},\mathbf{e_y}} = +1 \, (-1)$ for sites with even (odd) $x$-coordinate (green and red bullets, respectively). The links with phase $+1$ are depicted with red shaded ellipses and those with phase $-1$ are depicted with black crosses. c. Permitted local gauge-field configurations of the $G = 0$ sector of the Hilbert space. There are four possible configurations around a positron (I) and an electron (II) for $m/t = -\infty$; and 6 allowed configurations around charge neutral vacuum for $m/t = \infty$ (labels III and IV).

relations with the links:

$$\left[ \hat{E}_{\mathbf{j},\mathbf{e}_\mu}, \hat{U}_{\mathbf{k},\mathbf{e}_\nu} \right] = \delta_{\mathbf{j},\mathbf{k}} \delta_{\mu,\nu} \hat{U}_{\mathbf{j},\mathbf{e}_\mu}, \tag{1a}$$

$$\left[ \hat{E}_{\mathbf{j},\mathbf{e}_\mu}, \hat{U}^\dagger_{\mathbf{k},\mathbf{e}_\nu} \right] = -\delta_{\mathbf{j},\mathbf{k}} \delta_{\mu,\nu} \hat{U}^\dagger_{\mathbf{j},\mathbf{e}_\nu}. \tag{1b}$$

In the Wilson formulation of lattice gauge theories [58], the gauge field operators act on an infinite dimensional Hilbert space, and the operators $U$ and $U^\dagger$ at the same link commute. For quantum link models (QLMs) [59–61], the finite-dimensional Hilbert space requires the gauge fields to satisfy

$$\left[ \hat{U}_{\mathbf{j},\mathbf{e}_\mu}, \hat{U}^\dagger_{\mathbf{k},\mathbf{e}_\nu} \right] = \delta_{\mathbf{j},\mathbf{k}} \delta_{\mu,\nu} 2\hat{E}_{\mathbf{j},\mathbf{e}_\mu}. \tag{2}$$

The non-commuting nature of the gauge field operators has no adverse effects on the gauge invariance, since only Eqs. (1) are necessary to prove the local invariance of the Hamiltonian we will consider. Even more, it is precisely because of this property that the QLMs have a richer phase diagram than their Wilson counterparts. It can be shown that the Hilbert space of the link models can be scaled in a controlled fashion in order to reach the Wilson–Kogut–Susskind formulation [55, 56].

In the case of U(1) gauge theories, the link operators can be represented by quantum spin operators $\hat{\mathbf{S}} = (\hat{S}^x, \hat{S}^y, \hat{S}^z)$. Working in the $\hat{S}^z$-basis, we identify the electric flux operator as $\hat{E}_{\mathbf{j},\mathbf{e}_\mu} = \hat{S}^z_{\mathbf{j},\mathbf{e}_\mu}$. As sketched in Fig. 1a, the directions of arrows on the links indicate their $\hat{S}^z$ orientation. The upward (rightward) arrows on the vertical (horizontal) links denote an eigenvalue $+\frac{1}{2}$ of $\hat{S}^z$, and the downward (leftward) ones denote an eigenvalue of $-\frac{1}{2}$. The gauge-field operators raise and lower the flux at each link,

$$\hat{U}_{\mathbf{j},\mathbf{e}_\mu} = \hat{S}^x_{\mathbf{j},\mathbf{e}_\mu} + \mathrm{i}\hat{S}^y_{\mathbf{j},\mathbf{e}_\mu}, \tag{3a}$$

$$\hat{U}^\dagger_{\mathbf{j},\mathbf{e}_\mu} = \hat{S}^x_{\mathbf{j},\mathbf{e}_\mu} - \mathrm{i}\hat{S}^y_{\mathbf{j},\mathbf{e}_\mu}. \tag{3b}$$

In this paper, we will investigate the limit where the magnitude of the spin representation has the smallest possible value $|\hat{\mathbf{S}}| = S = \frac{1}{2}$, and therefore displays physics that is outside the scope of the Wilson formulation. This limit is particularly pertinent for gauge-theory implementations in quantum simulators that represent gauge fields with a two-dimensional Hilbert space [62–64]. As mentioned before, within the link formulation, one can consider spin representations with increasing $S$ to recover the Wilson–Kogut–Susskind limit. In that case, as is well known, the gauge-field operators become the corresponding raising and lowering operators operating in the infinite-dimensional local Hilbert space of a quantum rotor and recover the commutation of $U$ and $U^\dagger$ [65].

The QLM Hamiltonian with a dynamical quantum electromagnetic gauge field coupled to a single flavor of charged matter can be written as [66]

$$\begin{aligned}
\hat{H} = &-t \sum_{\mathbf{j},\mu=x,y} s_{\mathbf{j},\mathbf{e}_\mu}\left(\hat{\psi}^\dagger_{\mathbf{j}} \hat{U}_{\mathbf{j},\mathbf{e}_\mu} \hat{\psi}_{\mathbf{j}+\mathbf{e}_\mu} + \hat{\psi}^\dagger_{\mathbf{j}+\mathbf{e}_\mu} \hat{U}^\dagger_{\mathbf{j},\mathbf{e}_\mu} \hat{\psi}_{\mathbf{j}}\right) \\
&+ m \sum_{\mathbf{j}} s_{\mathbf{j}} \hat{\psi}^\dagger_{\mathbf{j}} \hat{\psi}_{\mathbf{j}} + \frac{g^2}{2} \sum_{\mathbf{j},\mu=x,y} \hat{E}^2_{\mathbf{j},\mathbf{e}_\mu} \\
&- J \sum_{\square} \left(\hat{U}_{\square} + \hat{U}^\dagger_{\square}\right).
\end{aligned} \tag{4}$$

In this formulation, the hopping and mass terms are staggered by adopting the Kogut–Susskind framework [58] in order to avoid the doubling problem in the conventional continuum limit [67]. The hopping of the fermions is staggered via a coefficient $s_{\mathbf{j},\mathbf{e}_\mu}$, which takes the direction-dependent values $s_{\mathbf{j},\mathbf{e}_x} = 1$ and $s_{\mathbf{j},\mathbf{e}_y} = (-1)^{j_x}$. This staggering factor explicitly breaks the lattice translational symmetry by a single lattice site, and doubles the Brillouin zone. If one takes the conventional continuum limit, this reduces the number of doublers [58]. The staggered mass term has the phase $s_{\mathbf{j}} = (-1)^{j_x+j_y}$ to correspond to a hole when the fermion is present on an even site ($s_{\mathbf{j}} = 1$) and an electron when the fermion is present on an odd site ($s_{\mathbf{j}} = -1$). For the gauge fields in the spin $S = \frac{1}{2}$ representation, the electric field energy term is a constant and can be ignored. Finally, magnetic interactions for the gauge fields are governed by the plaquette terms $\hat{U}_{\square}$ involving four-body interactions with $\mathbf{j}$ denoting the bottom left site of the plaquette:

$$\hat{U}_{\square} = \hat{U}_{\mathbf{j},\mathbf{e}_x} \hat{U}_{\mathbf{j}+\mathbf{e}_x,\mathbf{e}_y} \hat{U}^\dagger_{\mathbf{j}+\mathbf{e}_y,\mathbf{e}_x} \hat{U}^\dagger_{\mathbf{j},\mathbf{e}_y}. \tag{5}$$

Figure 1a shows a configuration of gauge links around different plaquettes. With a two-dimensional local Hilbert space, there are 16 possible states on a plaquette. The plaquette Hamiltonian acts nontrivially only on two of them, which have a clockwise or anticlockwise circulation of electric flux along the links of the plaquette. Such a plaquette is called flippable, such as the ones marked as 1, 2, 3, 4, and 5 in Fig. 1a. The plaquette term converts a clockwise-plaquette to an anticlockwise one, and vice-versa. Plaquettes that have no clear

orientation of fluxes are annihilated. We note that for a relativistic theory obtained at the traditional continuum limit, the couplings $g^2$ and $J$ are related with the lattice spacing $a$, for example $J \sim 1/(ag^2)$. In our case, however, since we want to explore the phase diagram for the different values of bare couplings, we do not assume this relation beforehand.

The Hamiltonian commutes with the Gauss's law operators at each site $\mathbf{j}$ defined as

$$\hat{G}_{\mathbf{j}} = \hat{\psi}_{\mathbf{j}}^\dagger \hat{\psi}_{\mathbf{j}} - \frac{1 - (-1)^{j_x + j_y}}{2} - \sum_\mu \left( \hat{E}_{\mathbf{j},\mathbf{e}_\mu} - \hat{E}_{\mathbf{j}-\mathbf{e}_\mu,\mathbf{e}_\mu} \right). \tag{6}$$

These generate the local U(1) transformation $\prod_{\mathbf{j}} \exp\left(-i\theta_{\mathbf{j}}\hat{G}_{\mathbf{j}}\right)$, under which the Hamiltonian (and hence the physical spectrum) remains invariant. Accordingly, one can choose a target superselection sector in which to study the phase diagram of the model. These superselection sectors are defined by the local charges $g_{\mathbf{j}}$, which are the eigenvalues of the Gauss law operators, $\hat{G}_{\mathbf{j}}|\Psi\rangle = g_{\mathbf{j}}|\Psi\rangle$. Following the common convention in particle physics, we choose the sector with $g_{\mathbf{j}} = 0$ for all $\mathbf{j}$. Physically, the vacuum is free of dynamical charges due to the staggered occupation of the fermions in the static limit (where $m/t \to \pm\infty$). Together with the global charge conjugation symmetry, this limits us to the case where the fermionic matter fills half of the spatial volume.

The constraints imposed by the local gauge symmetry fix the available gauge configurations for each fermionic occupation (the computational basis). In the limit of extreme values of the rest mass, namely $m/t = \pm\infty$ (where we have used the fermion hopping, $t$, to set the energy scale), the allowed gauge-field configurations are limited to 6 and 4 configurations per site, respectively, due to the difference in the local charge density, see Fig. 1c. These limiting values of $m$ are related to the quantum dimer model (QDM, $m/t = -\infty$) [68] and pure gauge quantum link model (QLM, $m/t = +\infty$) [48, 69]. Thus, this model can be used to probe the crossover behavior between these two paradigmatic models, and possible phase transitions associated with the change in the order caused by the changing constraints in the local degrees of freedom.

## 2.2 Global symmetries

The understanding of the global symmetries of the model is essential to decode its phase diagram as well as the nature of the symmetry-broken phases it realizes. These are the discrete point group symmetries such as the shift, lattice translation, reflection, and rotation symmetries. Note that the presence of the staggered phase factors $s_{\mathbf{j},\mathbf{e}_\mu}$ for the relativistic staggered fermions cause the lattice shifts by a single lattice spacing and the lattice translations to be different from each other, and to be defined according to the direction. Among the internal symmetries, only the discrete charge conjugation survives (as compared to the pure gauge theory), since the U(1) global winding-number symmetries for the pure gauge theory are no longer exact in the presence of dynamical matter fields. We describe below how the fermionic fields and the gauge fields transform under these symmetries.

- **Shift symmetry** $\mathcal{S}_k$ is the shift (translation) of the system by one lattice spacing in the $k$−th direction. This symmetry is the analog of continuum chiral symmetry of the Dirac fermions. As is well known [10], the Kogut-Susskind formulation has the different components of the Dirac fermions arranged on different sublattices. The usual chiral symmetry, which mixes the different components of the Dirac fermions, is therefore equivalent to shift symmetries of the staggered fermions. Lattice staggered fermions realize a smaller chiral symmetry than the original Dirac fermions. The discrete chiral symmetry

of the fermions in our model transform the fields as:

$$\mathcal{S}_x \hat{\psi}_{\mathbf{j}} = (-1)^{\mathbf{j}_y} \hat{\psi}_{\mathbf{j}+\mathbf{e}_x} \,, \qquad\qquad \mathcal{S}_y \hat{\psi}_{\mathbf{j}} = \hat{\psi}_{\mathbf{j}+\mathbf{e}_y} \,, \tag{7a}$$

$$\mathcal{S}_x \hat{U}_{\mathbf{j},\mathbf{e}_\mu} = \hat{U}_{\mathbf{j}+\mathbf{e}_x,\mathbf{e}_\mu} \,, \qquad\qquad \mathcal{S}_y \hat{U}_{\mathbf{j},\mathbf{e}_\mu} = \hat{U}_{\mathbf{j}+\mathbf{e}_y,\mathbf{e}_\mu} \,, \tag{7b}$$

$$\mathcal{S}_x \hat{E}_{\mathbf{j},\mathbf{e}_\mu} = \hat{E}_{\mathbf{j}+\mathbf{e}_x,\mathbf{e}_\mu} \,, \qquad\qquad \mathcal{S}_y \hat{E}_{\mathbf{j},\mathbf{e}_\mu} = \hat{E}_{\mathbf{j}+\mathbf{e}_y,\mathbf{e}_\mu} \,. \tag{7c}$$

The shift symmetry by two lattice spacings in the $k$−th direction is the ordinary **translation symmetry,** $T_k$ in the $k$−th direction. The conserved momentum takes values in $-\frac{\pi}{2a}$ and $\frac{\pi}{2a}$. Shifts by one lattice spacing in a single direction correspond to the discrete $\mathbb{Z}_2$ chiral symmetry, while shifts by one lattice spacing in each of the two directions is the discrete $\mathbb{Z}_2$ flavor symmetry. In the zero momentum sector, these shifts $\mathcal{S}_k$ form a $\mathbb{Z}_2 \times \mathbb{Z}_2$ group.

- **Charge conjugation** $C_k$ is an internal discrete symmetry that acts differently in different directions, like the shift symmetry. It is implemented as:

$$C_x \hat{\psi}_{\mathbf{j}} = (-1)^{\mathbf{j}_x} \hat{\psi}^\dagger_{\mathbf{j}+\mathbf{e}_x} \,, \qquad\qquad C_y \hat{\psi}_{\mathbf{j}} = \hat{\psi}^\dagger_{\mathbf{j}+\mathbf{e}_y} \,, \tag{8a}$$

$$C_x \hat{U}_{\mathbf{j},\mathbf{e}_\mu} = \hat{U}^\dagger_{\mathbf{j}+\mathbf{e}_x,\mathbf{e}_\mu} \,, \qquad\qquad C_y \hat{U}_{\mathbf{j},\mathbf{e}_\mu} = \hat{U}^\dagger_{\mathbf{j}+\mathbf{e}_y,\mathbf{e}_\mu} \,, \tag{8b}$$

$$C_x \hat{E}_{\mathbf{j},\mathbf{e}_\mu} = -\hat{E}_{\mathbf{j}+\mathbf{e}_x,\mathbf{e}_\mu} \,, \qquad\qquad C_y \hat{E}_{\mathbf{j},\mathbf{e}_\mu} = -\hat{E}_{\mathbf{j}+\mathbf{e}_y,\mathbf{e}_\mu} \,. \tag{8c}$$

It is interesting to note that the action of a single $C_k$ is a genuine charge conjugation transformation, while applying it twice gives rise to a lattice translation (when applied in the same direction) or a flavor transformation (when applied successively in the two directions).

- The **parity** transformation $\mathcal{P}$ acts the same way in both the directions:

$$\mathcal{P} \hat{\psi}_{\mathbf{j}} = \hat{\psi}_{-\mathbf{j}} \,, \tag{9a}$$

$$\mathcal{P} \hat{U}_{\mathbf{j},\mathbf{e}_\mu} = \hat{U}^\dagger_{-\mathbf{j}-\mathbf{e}_\mu,\mathbf{e}_\mu} \,, \tag{9b}$$

$$\mathcal{P} \hat{E}_{\mathbf{j},\mathbf{e}_\mu} = -\hat{E}_{-\mathbf{j}-\mathbf{e}_\mu,\mathbf{e}_\mu} \,. \tag{9c}$$

Note that in two spatial dimensions, the parity operation can be realized as a lattice reflection on one of the lattice axes. Defining a parity operation that flips both the axes can also be realized with a lattice rotation.

In addition, other lattice symmetries that we will not consider in detail in this article are rotation and reflection, which do not all commute with the above transformations.

Finally, let us remark that the winding-number operators in the $x$- and $y$-directions, defined as

$$\hat{W}_x = \frac{1}{L_x L_y} \sum_{\mathbf{j}} \hat{E}_{\mathbf{j},\mathbf{e}_y} \,, \tag{10a}$$

$$\hat{W}_y = \frac{1}{L_x L_y} \sum_{\mathbf{j}} \hat{E}_{\mathbf{j},\mathbf{e}_x} \,, \tag{10b}$$

generate a global $U(1) \times U(1)$ symmetry associated with the center of the gauge groups. However, in the presence of dynamical fermions, the winding numbers are not good global symmetries. Nevertheless their expectation values can indicate the ease with which global strings, or strings joining dynamical charges can be excited in the ground state, and thus indicate the confining or the deconfining nature of the system. For a discussion of our results on the winding numbers please see Appendix A.

# 3  Numerical Method

To calculate the ground state of the model, we use the infinite-size density matrix renormalization group technique (iDMRG) [20, 21, 23–25]. The system has a cylindrical geometry that is infinitely long along its axis ($x$-direction, open boundary conditions), and comprises $L_y = 4$ fermionic sites along its circumference ($y$-direction, periodic boundary conditions). An analysis of convergence of our results with respect to bond dimension is given in Appendix B, while a system-size analysis is provided in Appendix C. The local basis for the lattice sites is constructed by representing the gauge links with rishon fermions [17, 28, 29, 54, 70]. This formulation represents the two-dimensional Hilbert space of a gauge field on a bond between sites $\mathbf{j}$ and $\mathbf{j} + \mathbf{e}_\mu$ as the two positions of a single auxiliary fermionic particle that can reside either to the left or to the right of the link. Using the operators $\hat{c}^\dagger_{\mathbf{j},\mathbf{e}_\mu}$ and $\hat{c}^\dagger_{\mathbf{j}+\mathbf{e}_\mu,-\mathbf{e}_\mu}$ to represent creation operators of the rishons to the left and right side of the link, the action of the gauge field on the link is $\hat{U}_{\mathbf{j},\mathbf{e}_\mu} = \hat{c}_{\mathbf{j},\mathbf{e}_\mu} \hat{c}^\dagger_{\mathbf{j}+\mathbf{e}_\mu,-\mathbf{e}_\mu}$. The electric-flux operator in terms of rishons is given by $\hat{E}_{\mathbf{j},\mathbf{e}_\mu} = (\hat{n}_{\mathbf{j}+\mathbf{e}_\mu,-\mathbf{e}_\mu} - \hat{n}_{\mathbf{j},\mathbf{e}_\mu})/2$, with $\hat{n}_{\mathbf{j},\mathbf{e}_\mu} = \hat{c}^\dagger_{\mathbf{j},\mathbf{e}_\mu} \hat{c}_{\mathbf{j},\mathbf{e}_\mu}$ denoting the rishon number operator at the respective rishon sites located to the left and right of the link directed from $\mathbf{j}$ to $\mathbf{j} + \mathbf{e}_\mu$.

The creation and annihilation operators for rishon fermions obey the usual anti-commutation relations $\{\hat{c}_{\mathbf{j},\mathbf{e}_\mu}, \hat{c}_{\mathbf{k},\mathbf{e}_\nu}\} = 0$ and $\{\hat{c}^\dagger_{\mathbf{j},\mathbf{e}_\mu}, \hat{c}_{\mathbf{k},\mathbf{e}_\nu}\} = \delta_{\mathbf{j},\mathbf{k}}\delta_{\mu,\nu}$. The $S = \frac{1}{2}$ spin algebra of each bond is embedded in the sector of the Hilbert space of two rishon fermions with the constraint

$$\left( \hat{c}^\dagger_{\mathbf{j},\mathbf{e}_\mu} \hat{c}_{\mathbf{j},\mathbf{e}_\mu} + \hat{c}^\dagger_{\mathbf{j}+\mathbf{e}_\mu,-\mathbf{e}_\mu} \hat{c}_{\mathbf{j}+\mathbf{e}_\mu,-\mathbf{e}_\mu} \right) |\Psi\rangle = |\Psi\rangle \,. \tag{11}$$

This constraint can be associated with an Abelian link gauge symmetry that keeps the rishon number at each link fixed. The basis state of up and down spins is represented by the rishon fermions as

$$|\!\uparrow\rangle_{\mathbf{j},\mathbf{e}_\mu} = |0,1\rangle = \hat{c}^\dagger_{\mathbf{j}+\mathbf{e}_\mu,-\mathbf{e}_\mu} |0\rangle_{\mathbf{j},\mathbf{e}_\mu} |0\rangle_{\mathbf{j}+\mathbf{e}_\mu,-\mathbf{e}_\mu} \,, \tag{12a}$$

$$|\!\downarrow\rangle_{\mathbf{j},\mathbf{e}_\mu} = -|1,0\rangle = -\hat{c}^\dagger_{\mathbf{j},\mathbf{e}_\mu} |0\rangle_{\mathbf{j},\mathbf{e}_\mu} |0\rangle_{\mathbf{j}+\mathbf{e}_\mu,-\mathbf{e}_\mu} \,, \tag{12b}$$

where the negative sign is necessary for recovering the relation $U |\!\downarrow\rangle = |\!\uparrow\rangle$.

In this formulation, the configuration of the gauge field around a charged fermion can be accessed by the occupation status of the closest rishon fermions. Gauss' law around the site, therefore, becomes a constraint on the total number of fermions at and around the site,

$$g_{\mathbf{j}} = n_{\mathbf{j}} + n_{\mathbf{j},\mathbf{e}_x} + n_{\mathbf{j},-\mathbf{e}_x} + n_{\mathbf{j},\mathbf{e}_y} + n_{\mathbf{j},-\mathbf{e}_y}$$
$$- 2 - \frac{(-1)^{\mathbf{j}_x+\mathbf{j}_y+1} + 1}{2} \,, \tag{13}$$

where $n_{\mathbf{j}}$ is the eigenvalue of the fermionic number operator $\hat{n}_{\mathbf{j}} = \hat{\psi}^\dagger_{\mathbf{j}} \hat{\psi}_{\mathbf{j}}$, and $n_{\mathbf{j},\mathbf{e}_\mu}$ is the eigenvalue of the rishon number operator $\hat{n}_{\mathbf{j},\mathbf{e}_\mu} = \hat{c}^\dagger_{\mathbf{j},\mathbf{e}_\mu} \hat{c}_{\mathbf{j},\mathbf{e}_\mu}$. It is more convenient to define the local basis as the combined basis of the state at and around a fermionic site. This results in a representation of the local basis in the form

$$\left| \begin{array}{ccc} & n_{\mathbf{j},\mathbf{e}_y} & \\ n_{\mathbf{j},-\mathbf{e}_x} & n_{\mathbf{j}} & n_{\mathbf{j},\mathbf{e}_x} \\ & n_{\mathbf{j},-\mathbf{e}_y} & \end{array} \right\rangle = (-1)^{n_{\mathbf{j},\mathbf{e}_x}+n_{\mathbf{j},\mathbf{e}_y}} |n_{\mathbf{j}}\rangle |n_{\mathbf{j},-\mathbf{e}_x}\rangle |n_{\mathbf{j},-\mathbf{e}_y}\rangle |n_{\mathbf{j},\mathbf{e}_x}\rangle |n_{\mathbf{j},\mathbf{e}_y}\rangle \,. \tag{14}$$

Although the basis state obeys Gauss' law, it does not immediately fulfill the constraint (11) for the rishon fermions. For the simulation using matrix product states (MPS), the constraint

is imposed in two ways. First, in order to satisfy the constraint in the $y$-direction, the local basis of an MPS is taken to be the basis formed by the tensor products of the $L_y$ local bases of a rung of the cylinder. This way, any local basis that does not obey the constraint can be rejected from the list of local bases that MPS uses for simulating the ground state.

For the links in the $x$-direction, we have introduced two auxiliary charges, $q^{\text{even}}$ and $q^{\text{odd}}$ [28, 70]. The value of the charges at each site is given by $q_{\mathbf{j}}^{\lambda} = 2n_{\mathbf{j}, \pm \mathbf{e}_x} - 1$ where $\lambda$ is either even or odd. Here, the sign $\pm$ is fixed to $+$ when $\lambda$ is even (odd) and $\mathbf{j}$ is in the even (odd) sublattice, and $-$ otherwise. By imposing the constraint $\sum_{\mathbf{j}} q_{\mathbf{j}}^{\lambda} = 0$ for each of the sublattices $\lambda$, the constraint (11) is satisfied. As these auxiliary charges are good quantum numbers, they can readily be implemented in the framework of matrix product states and operators as global U(1) symmetries [26, 71–75].

We have also benchmarked the iDMRG calculation with an exact diagonalization (ED) calculation at the extreme parameter limits ($m/t = \pm\infty$). In this limit, since the occupation number of the fermions gets frozen, the number of allowed states in the Hilbert space is smaller, thereby facilitating the ED calculations. The comparison is quantitatively discussed in Appendix D.

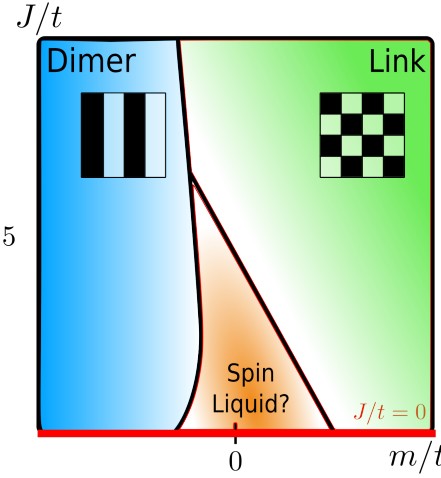

Figure 2: Schematic of the phase diagram. As explained in the text, the $J/t = 0$ and $J/t > 0$ regions (corresponding to the absence, respectively, presence of the plaquette interaction) are discussed separately. For small and finite $J/t$, we conjecture the existence of a spin-liquid like phase that exhibits much slower decay of correlations of the fermion number and the plaquette flippability as compared to the other regions. This region shrinks as $J/t$ is increases. For large $J/t$, a first order transition is likely between the quantum link-like and the quantum dimer-like phases. Colors are only for illustrative purposes.

## 4 Phase Diagram

In this section, we present the numerically extracted phase diagram of the model given by the Hamiltonian in Eq. (4) and Gauss' law as defined in Eq. (6). We start first in the limit of vanishing strength of the plaquette terms $J/t = 0$ and include the discussion of finite values $J/t > 0$ as a second step. This division is motivated by the fact that the presence or absence of the plaquette terms changes the symmetry broken by the ground state. For each of the two cases, we discuss the phase diagram as the parameter $m/t$ is changed from $-\infty$ to $+\infty$. Figure 2 shows a schematic phase diagram indicating the different phases that exist in this

model, and the lines of possible phase transitions between them.

The limits of extreme values of the rest mass are well known models, QDM and pure gauge QLM, respectively. For large positive values of the mass, one can construct an effective field theory, analogous to the heavy quark effective theory (HQET) [76,77] as a deviation from the ground state of the pure gauge theory. However, we emphasize that such a construction is intrinsically nonperturbative for $J/t > 0$, since the ground state of the pure gauge theory is strongly interacting. For small $J/t$, the construction of a perturbative expansion is straightforward. Since we have cross-checked our numerical results with exact diagonalization, we bypass such analytic treatments (see Appendix D).

## 4.1 $J/t = 0$

We start by focusing on the model in the absence of plaquette terms, i.e., by fixing $J/t = 0$. To understand the phase diagram, we first examine the two extreme limits of the large-mass regimes, $m/t = \pm\infty$. In these limits, the Hamiltonian becomes diagonal in the chosen computational basis (see Fig. 1c), and the ground states within the target sector are the lowest-energy eigenstates that satisfy Gauss' law. Since the large mass freezes the positions of the fermions, they act as background charges in the problem. The Hilbert space is exponentially reduced as compared to the regime of finite $m$, which enables us to cross-check our iDMRG results with ED.

In the $m/t = +\infty$ limit, the fermions occupy odd sites, while holes occupy even sites. We can identify this as the charge-neutral vacuum, which then naturally realizes the pure gauge QLM with spin length $S = \frac{1}{2}$ [17]. This implies that six configurations of gauge fields are acceptable as shown in Fig. 1c (III and IV). In the opposite limit of $m/t = -\infty$, the occupation of the fermions switches from the odd to the even sites, and thus the vacuum now is proliferated with positive and negative charges distributed in a staggered fashion, which we will henceforth refer to as "positrons" and "electrons". Since the fermions are static in this limit, so is the charge distribution, and thus we naturally recover the Hilbert space of the QDM (on the square lattice) [48,69]. As shown in Fig. 1c (I and II), the presence (absence) of the particle (hole) places additional constraints on the gauge fields, and only four local configurations for the gauge field are allowed.

In these limiting cases, the hopping term of the Hamiltonian effectively vanishes, as fermions are not allowed to move between even and odd sites. Hence, the only term that we need to consider is the plaquette term. When $J/t = 0$, this term is absent, and so there is no mechanism that restricts the configuration of the gauge field other than Gauss' law. Thus, we expect the model to behave differently for $J/t = 0$ and $J/t \neq 0$.

For $J/t = 0$ (still with $m/t = \pm\infty$), the situation is analogous to the infinite-temperature limit of the pure gauge QLM, respectively, the QDM. In this case, due to the vanishingly small imaginary time $\beta \to 0$, the partition function $Z = \text{Tr}[\exp(-\beta\hat{H})\hat{\mathbb{P}}_G]$ —with $\hat{\mathbb{P}}_G$ the projector onto the target sector—weighs all states allowed by the corresponding Gauss law equally, and the fermions drop out of the problem. Gauge fields do not break any lattice symmetries except the center symmetries, and naively one expects the ground state to have winding strings. However, our results (see Appendix A) indicate the absence of such strings in the ground state. This region is indicated with a red bold line in the schematic phase diagram in Fig. 2.

In the opposite limit of vanishing bare mass, $m/t = 0$, the fermions are free to hop, and the physics is that of a strongly correlated system of fermions and gauge fields. It is interesting to ask if the system in this region generates a correlation length dynamically. However, as we will discuss, our results point to the existence of a liquid-like phase with no broken symmetries.

It is possible to achieve a quantitative characterization of the occurring phases using various order parameters that are sensitive to appropriate symmetry breakings. In the limit of $m/t = -\infty$, the ground state has a broken shift (chiral) symmetry $\mathcal{S}_x$ and $\mathcal{S}_y$ both along the

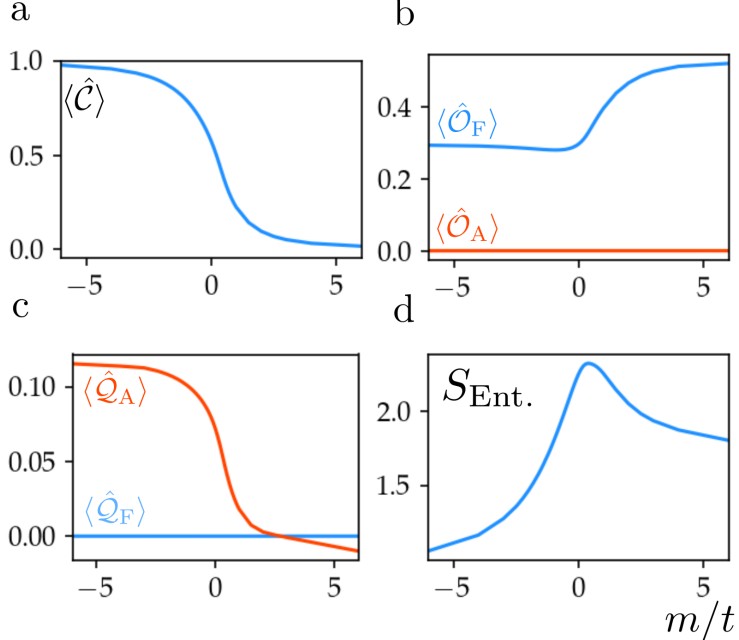

Figure 3: Physical observables for $J/t = 0$: a. Chiral condensate $\langle \hat{\mathcal{C}} \rangle$, b. the plaquette operators $\langle \hat{\mathcal{O}}_\mathrm{F} \rangle$, $\langle \hat{\mathcal{O}}_\mathrm{A} \rangle$ which detect the flippability of the plaquettes, c. plaquette operators $\langle \hat{\mathcal{Q}}_\mathrm{F} \rangle$, $\langle \hat{\mathcal{Q}}_\mathrm{A} \rangle$ which are sensitive to the (anti) clockwise ordering of the plaquettes, d. Entanglement entropy of a bipartition along the $y$-direction. As explained in the text, the expectation value of the charge conjugation operator indicates that the chiral symmetry breaking dissolves as $m/t$ is tuned from $-\infty$ to $\infty$. Simultaneously, the expectation values of the plaquette operators also show the restoration of the shift symmetry as we move into the QLM-like phase (at $m/t > 0$). The peak of the entanglement entropy could point to the existence of a disordered phase, with no broken symmetries, such that the ground state gets contributions from a large number of basis states.

$x$- and $y$-directions of the fermion occupation. Even sites are occupied, corresponding to the presence of a charged fermion, and leading to the additional breaking of the charge conjugation symmetry $C_k$ in both directions. The lattice translation $T_k$ symmetry defined as a shift by two lattice spacings remains intact. The order parameter that characterizes this phase is the chiral condensate

$$\hat{\mathcal{C}} = \frac{1}{L_x L_y} \sum_{\mathbf{j}} (-1)^{\mathbf{j}_x + \mathbf{j}_y} \left[ \hat{n}_{\mathbf{j}} - \frac{1 - (-1)^{\mathbf{j}_x + \mathbf{j}_y}}{2} \right], \tag{15}$$

which is shown in Fig. 3a as a function of $m/t$. The expression within the sum is just the modulus of the electric charge at site $\mathbf{j}$. The order parameter smoothly approaches 0 as $m/t$ approaches $+\infty$ with a sharp drop around $m/t = 0$. As the sign of the mass changes, the dominant configuration of the matter fermions goes from QDM-like to pure gauge QLM-like vacuum. This order parameter is only sensitive to the fermionic sector of the theory, and is an indicator of when the vacuum is unstable to charge–anti-charge fluctuations.

The ordering of the gauge field can be more subtle. While the disordered phase of the fermions (the regime of small $m/t$, where the fermions hop freely destroying any energetically favored order) also tends to disorder the gauge fields, the presence or absence of the $J/t$ term influences the ordering in the gauge field. This is somewhat analogous to the phenomenon

of 'order by disorder' first introduced in [78]. We introduce the following local operators to identify the correlations in the gauge fields:

$$\hat{\mathcal{O}}_{\mathbf{j}} = \sum_{\eta=\pm} \hat{P}^{\eta}_{\mathbf{j},\mathbf{e}_x} \hat{P}^{\eta}_{\mathbf{j}+\mathbf{e}_x,\mathbf{e}_y} \hat{P}^{\bar{\eta}}_{\mathbf{j}+\mathbf{e}_y,\mathbf{e}_x} \hat{P}^{\bar{\eta}}_{\mathbf{j},\mathbf{e}_y}, \tag{16a}$$

$$\hat{\mathcal{Q}}_{\mathbf{j}} = \sum_{\eta=\pm} \eta \hat{P}^{\eta}_{\mathbf{j},\mathbf{e}_x} \hat{P}^{\eta}_{\mathbf{j}+\mathbf{e}_x,\mathbf{e}_y} \hat{P}^{\bar{\eta}}_{\mathbf{j}+\mathbf{e}_y,\mathbf{e}_x} \hat{P}^{\bar{\eta}}_{\mathbf{j},\mathbf{e}_y}, \tag{16b}$$

where $\hat{P}^{\pm}_{\mathbf{j},\mathbf{e}_x} = \frac{1}{2} \pm \hat{E}_{\mathbf{j},\mathbf{e}_x}$ and $\bar{\eta}$ denotes the opposite sign of $\eta$. Individual plaquettes can be either flippable clockwise or anticlockwise, or nonflippable, such that the two operators in Eqs. (16) are sufficient to distinguish all these possibilities. $\hat{\mathcal{O}}_{\mathbf{j}}$ counts a flippable plaquette irrespective of its orientation, and $\hat{\mathcal{Q}}_{\mathbf{j}}$ is sensitive to the orientation, giving opposite signs for plaquettes that are flippable in the clockwise and anticlockwise directions. Using these local operators, we can design order parameters (normalized with the lattice volume $V = L_x L_y$) to detect the different kinds of ordering of the gauge fields,

$$\hat{\mathcal{O}}_{\mathrm{F}} = \frac{1}{V} \sum_{\mathbf{j}} \hat{\mathcal{O}}_{\mathbf{j}}, \quad \hat{\mathcal{O}}_{\mathrm{A}} = \frac{1}{V} \sum_{\mathbf{j}} (-1)^{\mathbf{j}} \hat{\mathcal{O}}_{\mathbf{j}}, \tag{17a}$$

$$\hat{\mathcal{Q}}_{\mathrm{F}} = \frac{1}{V} \sum_{\mathbf{j}} \hat{\mathcal{Q}}_{\mathbf{j}}, \quad \hat{\mathcal{Q}}_{\mathrm{A}} = \frac{1}{V} \sum_{\mathbf{j}} (-1)^{\mathbf{j}} \hat{\mathcal{Q}}_{\mathbf{j}}. \tag{17b}$$

Physically, these operators are analogs of the uniform and the staggered magnetizations commonly used to diagnose orderings in the context of spin systems. For example, consider the example when all plaquettes are flippable. This can only happen when the even and odd plaquettes are (all) flippable in opposite orientations. Therefore, $\langle \hat{\mathcal{O}}_{\mathrm{F}} \rangle \sim 1$, while $\langle \hat{\mathcal{O}}_{\mathrm{A}} \rangle \sim 0$. Similarly, $\langle \hat{\mathcal{Q}}_{\mathrm{F}} \rangle \sim 0$, but $\langle \hat{\mathcal{Q}}_{\mathrm{A}} \rangle \sim 1$. This scenario can occur in the pure gauge QLM due to an additional coupling, but not in our model.

These considerations help us understand the different numerically obtained scenarios realized by the present model, where we still focus for the moment on the $J/t = 0$ case (see Fig. 3). Generically, we will always get expectation values smaller than the idealized case discussed above due to the presence of dynamical fermions, which encourages disorder in the plaquettes. For $m/t = -\infty$, the even and the odd plaquettes have identical expectation values such that $\langle \hat{\mathcal{O}}_{\mathrm{A}} \rangle = 0$, while $\langle \hat{\mathcal{O}}_{\mathrm{F}} \rangle \approx 0.3$ (see Fig 3b), indicating that both the odd and the even plaquettes are coherently flippable, but approximately two-thirds of the plaquettes are non-flippable. In the infinite volume limit in both spatial directions, this number is known to be $\approx 0.5$ [68]. At the same time, $\langle \hat{\mathcal{Q}}_{\mathrm{F}} \rangle = 0$, $\langle \hat{\mathcal{Q}}_{\mathrm{A}} \rangle \approx 0.12$ (see Fig 3c) indicating that the flippable plaquettes on the even and the odd lattices have different orientations. Interestingly, this picture remains valid all the way to small values of $m/t$. For $m/t \to \infty$, every second plaquette (both among the odd, and among the even ones) becomes flippable, such that $\langle \hat{\mathcal{O}}_{\mathrm{A}} \rangle = 0$, $\langle \hat{\mathcal{O}}_{\mathrm{F}} \rangle \approx 0.5$ (see Fig 3c). Among both the even and the odd plaquettes, there are equal contributions from both clockwise and anticlockwise flippable plaquettes as evidenced by $\langle \hat{\mathcal{Q}}_{\mathrm{F}} \rangle \approx \langle \hat{\mathcal{Q}}_{\mathrm{A}} \rangle \approx 0$ (see Fig 3b). The expectation values of $\langle \hat{\mathcal{Q}}_{\mathrm{F}} \rangle$ suggest that while the shift symmetry is broken at $m/t \to -\infty$, it gets restored in the opposite limit of $m/t \to \infty$.

In addition, we find that the bipartite entanglement entropy increases as $m/t$ approaches 0 from $-\infty$, since the fermions are no longer quenched. However, beyond $m/t = 0$, the mobility of the fermions again decreases and there is an ordering in the gauge fields for large $m/t$. The peak of the bipartite entanglement entropy in Fig. 3d occurs at $m/t \sim 0$. Our results are in qualitative agreement with the ones observed in [51] for the parameter regime where both our Hamiltonians have the same form. We share the suggestion, also offered there, indicating

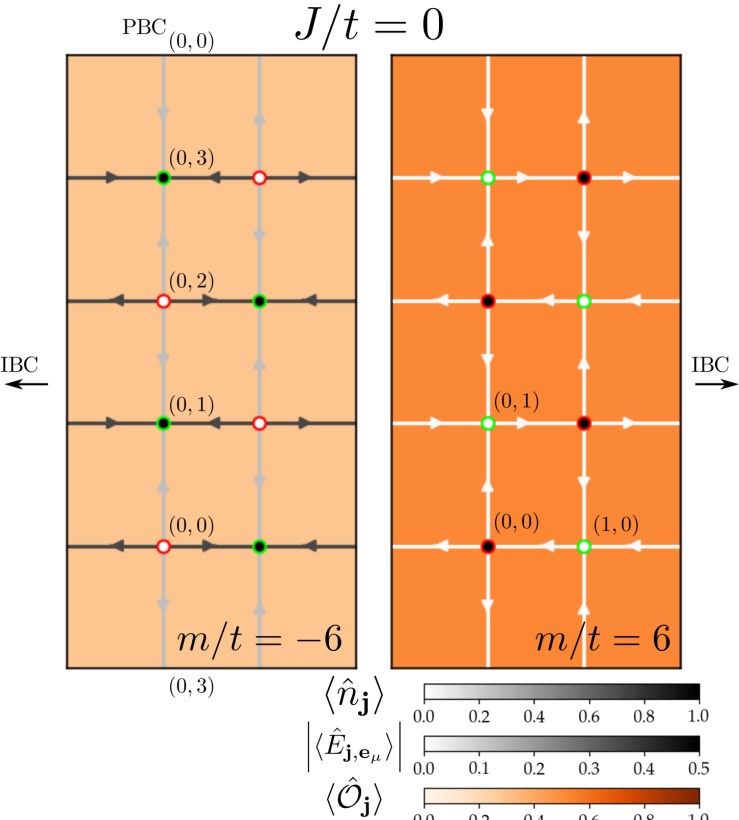

Figure 4: Electric flux and fermion number profiles at $m/t = \pm 6$. The QDM-like (left) and the QLM-like (right) patterns for the expectation values of the electric flux and the fermion number operator are clearly visible. The vacuum is proliferated with "electrons" and "positrons" and the electric flux breaks the shift symmetry in the QDM-like phase. The expectation value of the horizontal links is 0.36 and that of the vertical link is 0.12, which means that one out of three single plaquettes is flippable, consistent with $\langle \hat{\mathcal{O}}_F \rangle$. In the pure gauge QLM-like phase, the shift symmetry is restored and the staggered occupation of the fermions gives rise to the charge-neutral vacuum. The white color of the links denotes expectation values close to zero indicating that the shift symmetry is no longer broken. The plaquettes between $\mathbf{j}_y = 0$ and $\mathbf{j}_y = L_y - 1$ are duplicated at the top and the bottom to clearly illustrate the connections in the periodic boundary condition imposed on the $y$-direction.

a possible existence of a liquid-like phase in this regime. We will elaborate on this point more in the next section when we consider relevant correlation functions at $J/t > 0$.

Before going to discuss the case of $J/t > 0$, it is useful to look at the electric flux and the occupation number profiles, as shown in Fig. 4. For $m/t = -6$, the expectation value of the electric flux shows a staggering in both the $x$ and the $y$-directions, indicating the breaking of the shift symmetry. In contrast, the electric flux lines have vanishing expectation values for $m/t = 6$, which is the pure gauge QLM limit, confirming our previous observation that the shift symmetry is indeed restored for this regime.

## 4.2 $J/t > 0$

For a finite value of $J/t$, the phase diagram is more intricate since the plaquette term attempts to enforce its own order, which does not commute with the one established by the fermions.

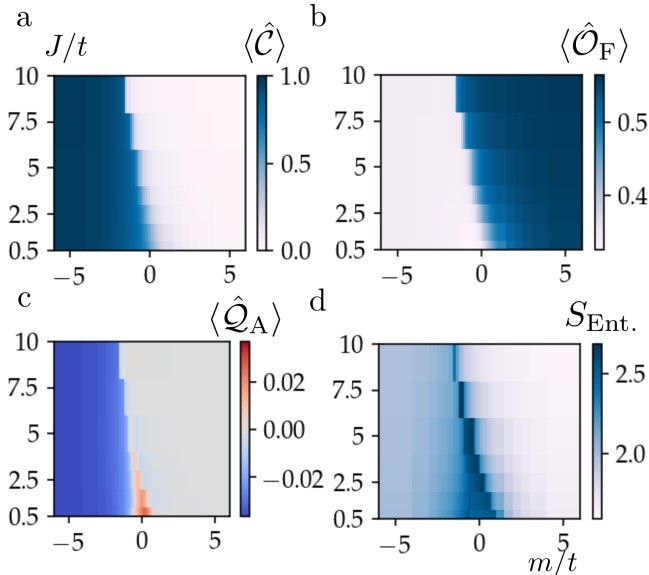

Figure 5: Physical observables for $J/t > 0$: a. Chiral condensate $\langle \hat{\mathcal{C}} \rangle$, b. plaquette operator $\langle \hat{\mathcal{O}}_\text{F} \rangle$, which is sensitive to the flippable plaquettes, c. plaquette operator $\langle \hat{\mathcal{Q}}_\text{A} \rangle$, sensitive to the (anti) clockwise ordering of the plaquettes, while $\langle \hat{\mathcal{Q}}_\text{F} \rangle = 0$, d. Entanglement entropy of a bipartition along $y$-direction. For small $J/t$ or order 1, there exists a third order between quantum dimer and quantum link models, which is associated with large bipartite entropy and inversion in the sign in $\langle \hat{\mathcal{O}}_F \rangle$.

This is also the regime beyond what has been considered in recent studies of a similar model [51, 53]. We start by examining the expectation values of the operators already introduced in the previous section, and then introduce correlation functions to accurately identify the physics of the ground state.

Let us first consider the behavior of the fermions, for which we refer to Fig. 5. The expectation value of the chiral condensate Eq. (15), $\langle \hat{\mathcal{C}} \rangle$, as shown in Fig. 5a, shows a similar behavior to the case of $J/t = 0$. However, the transition in $\langle \hat{\mathcal{C}} \rangle$ happens at smaller (absolute) values of $m/t$ for positive $J/t$ than at $J/t = 0$. Moreover the change from chiral-symmetry broken to restored phase becomes sharper with increasing $J/t$. To make sense of this behavior, we note that larger values of $J/t$ is consistent with the plaquette dynamics dominating the fermion hopping. The effect of the fermion mass term in this limit is only to change the vacuum. Therefore, depending on the value of $m/t$ the model is either in a QDM-like phase, or in a pure gauge QLM-like phase with a sharp transition separating them.

The gauge fields also change their ordering once $J/t$ is non-vanishing, and plaquette observables shown in Fig. 5b and c also show a sharper change with increasing $J/t$ as $m/t$ is tuned. In the QDM-like phase (for $m/t \to -\infty$), $\langle \hat{\mathcal{O}}_\text{A} \rangle = 0, \langle \hat{\mathcal{O}}_\text{F} \rangle \approx 0.35$, relatively independently of $J/t$. This value is about 15 % larger than in the case of $J/t = 0$. For all values of $J/t$, we find $\langle \hat{\mathcal{Q}}_\text{F} \rangle = 0$, indicating that spread over the lattice there must be as many plaquettes flippable clockwise as anticlockwise. However, $\langle \hat{\mathcal{Q}}_\text{A} \rangle$ shows a stronger dependence as a function of $J/t$ ranging between 0.12 when $J/t = 0$ to 0.04 for $J/t = 1.0$, above which it remains relatively unchanged. Taken together, both these observations seem to imply that the introduction of the plaquette term tends to make more plaquettes flippable, over and above the order decided by the fermion mass. However, if $\mathbf{j}$ and $\mathbf{j}'$ are neighboring plaquettes, and the plaquette $\mathbf{j}'$ is oscillating between the two orientations (as the plaquette term would favor), then the plaquette at $\mathbf{j}$ is never flippable. Thus, one of the sublattices supports an order that fluctuates between the two orientations, resulting in a decrease of $\langle \hat{\mathcal{Q}}_\text{A} \rangle$, while increasing



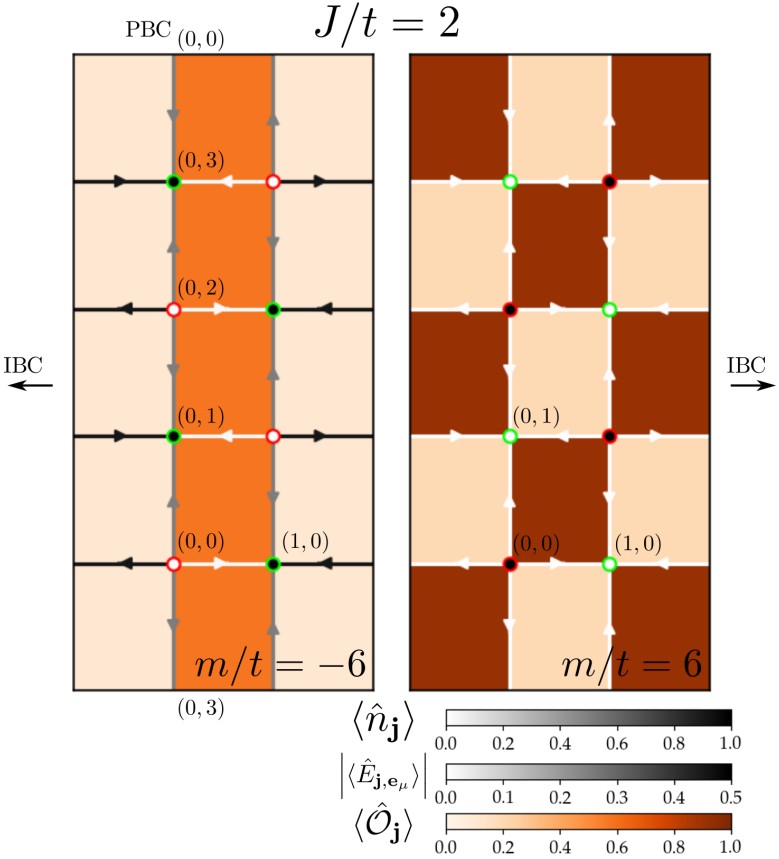

Figure 6: Electric flux and fermion number profiles for non-zero $J/t$ in both the QDM-like and the pure gauge QLM-like phase respectively. Darker color indicates an increased flippability. In the QDM-like phase there is a strip of plaquettes more flippable than their neighbors, the analog of the columnar phase(s). The pure gauge QLM-like phase shows spontaneous translation symmetry breaking with the plaquette term imposing a resonating pattern of flippable plaquettes on a sublattice. Moreover, this breaking is spontaneous for finite coupling $J/t$, similar to an order-by-disorder transition. The plaquettes between $\mathbf{j}_y = 0$ and $\mathbf{j}_y = L_y - 1$ are duplicated at the top and the bottom to clearly illustrate the connections in the periodic boundary condition imposed on the $y$-direction.

$\langle \hat{\mathcal{O}}_{\mathrm{F}} \rangle$.

As $m/t$ is increased, this scenario changes for a narrow region around $m/t \sim 0$, where the flippability drops, indicating that the fermion hops become more pronounced and play a role disordering the gauge fields, which we further discuss in the next paragraph. For positive $m/t$, as we approach the pure gauge QLM-like charge-neutral vacuum, the plaquette term favors the disordering, but more plaquettes are available on which it can establish the order that fluctuates the orientations. This is evidenced by the expectation values $\langle \hat{\mathcal{O}}_{\mathrm{A}} \rangle \approx 0.35$, $\langle \hat{\mathcal{O}}_{\mathrm{F}} \rangle \approx 0.55$ (see Fig. 5c), and breaks the shift symmetry spontaneously. Moreover, the ordering in our iDMRG results sometimes occurs on the even sublattice and sometimes on the odd one, clearly indicating the spontaneous nature of the symmetry breaking. Further, the values $\langle \hat{\mathcal{Q}}_{\mathrm{A}} \rangle \approx \langle \hat{\mathcal{Q}}_{\mathrm{F}} \rangle \approx 0$ (see Fig. 5b) are completely consistent with this scenario. Note the small island for small $J/t$ and $m/t \sim 0$ where $\langle \hat{\mathcal{Q}}_{\mathrm{A}} \rangle$ changes sign and is very small, along with all other plaquette observables. The entanglement entropy $S_{\mathrm{Ent}}$, displayed in Fig. 5d, is also peaked in this region.

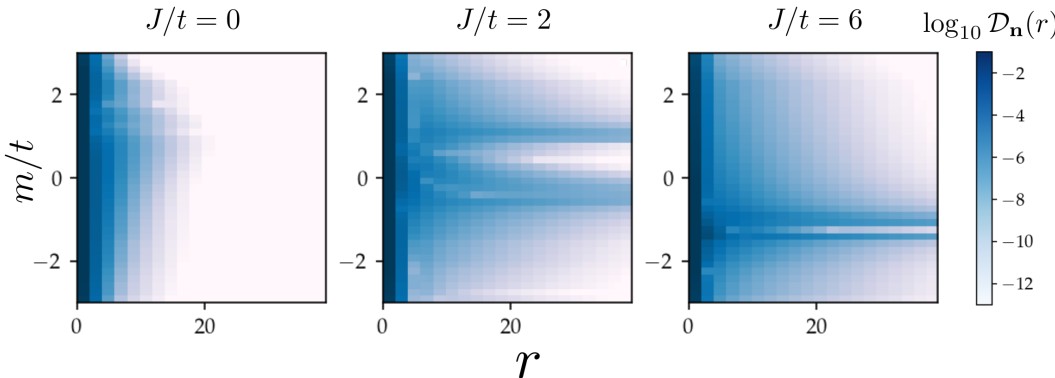

Figure 7: Density-density correlations $\mathcal{D}_\mathbf{n}(r)$ as a function of distance $r$ from $\mathbf{j} = (0,0)$ plotted for the values of $J/t = 0, 2, 6$. There is no long range order in the fermion number operator for $J/t = 0$, but for $J/t = 2$ there are two values of $m/t$ at which the correlation length becomes large, which enclose a region of smaller correlation length. The width of this region shrinks and fuses for large enough $J/t$.

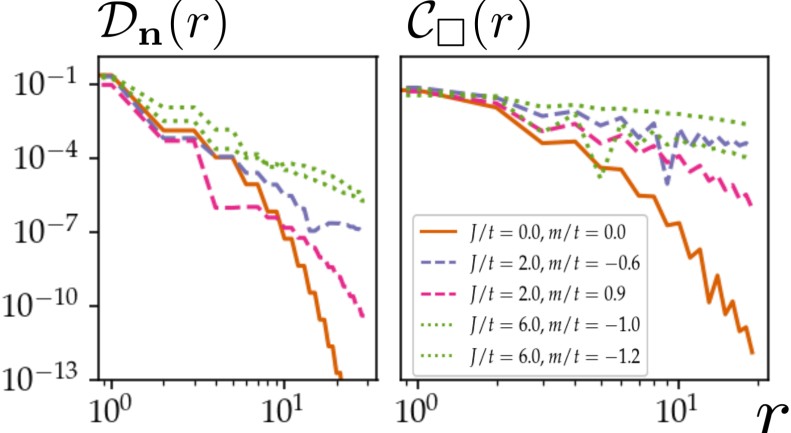

Figure 8: Fermionic (left) and plaquette correlation (right) functions as a function of distance $r$ for $J/t = 0$ and $m/t = 0$ (solid), $J/t = 2$ and $m/t = -0.6, 1.0$ (dashed), and $J/t = 6$ and $m/t = -1.0, -1.2$ (dotted). Regions where the gauge fields are correlated over large distances are at the boundary of the phases, between the QDM-like and liquid-like, or the QLM-like and liquid-like phase.

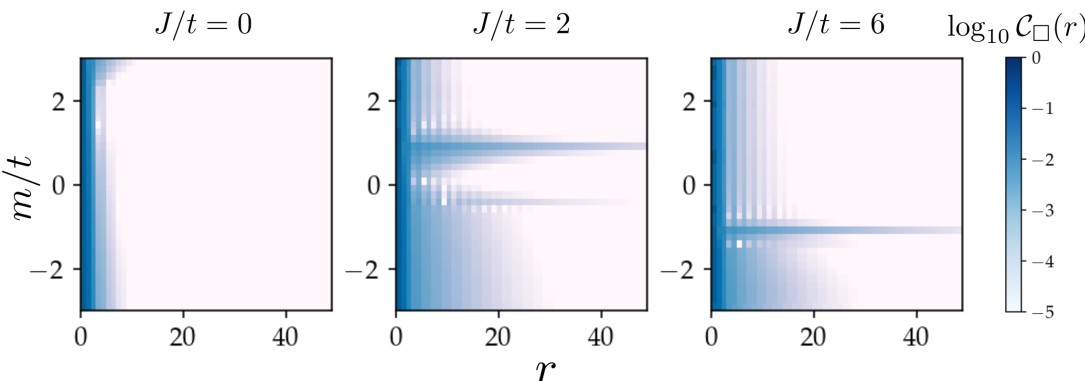

Figure 9: Plaquette-plaquette correlations $C_\square(r)$ as a function of distance from $\mathbf{j} = (0,0)$ plotted for the values of $J/t = 0, 2, 6$ and a range of $m/t$. As in the case of the fermion number correlators, a finite $J/t$ causes a larger correlation between the gauge fields, more notably in the QDM-like phase.

Moreover, we study the symmetry-breaking patterns in the expectation values of the fermion numbers and the electric fluxes as shown in Fig. 6 for $J/t = 2$. The former is rather similar to $J/t = 0$, while the latter reveals an interesting difference. For the QDM-like phase, as in the case of $J/t = 0$, the electric fluxes show a breaking of shift symmetry, but this is further manifested in the flippability of the dimers. Columnar strips of flippable plaquettes are created (represented by the darker color), just as in the pure QDM, which are flanked by strips of less flippable plaquettes (in lighter shade). The pure gauge QLM-like phase, on the other hand, undergoes an order-by-disorder transition [78] for any finite coupling $J/t$, best exhibited as a change in the flippability of the plaquettes. The resulting order created is best visualized through the plaquettes becoming flippable either on the odd or on the even sublattice, again represented in Fig. 6 as dark and light shades.

We investigate the correlation functions for the fermion occupation number as well as the plaquette flip operator to understand the nature of this region better, which are respectively defined as

$$\mathcal{D}_{\mathbf{n}}(r) = \langle \hat{n}_{\mathbf{j}} \hat{n}_{\mathbf{j}+r\mathbf{e}_x} \rangle - \langle \hat{n}_{\mathbf{j}} \rangle \langle \hat{n}_{\mathbf{j}+r\mathbf{e}_x} \rangle \,, \tag{18a}$$

$$\mathcal{C}_{\square}(r) = \langle \hat{P}_{\square,\mathbf{j}} \hat{P}_{\square,\mathbf{j}+r\mathbf{e}_x} \rangle - \langle \hat{P}_{\square,\mathbf{j}} \rangle \langle \hat{P}_{\square,\mathbf{j}+r\mathbf{e}_x} \rangle \,, \tag{18b}$$

where $\hat{P}_{\square,\mathbf{j}} = \hat{P}^+_{\mathbf{j},\mathbf{e}_x} \hat{P}^+_{\mathbf{j}+\mathbf{e}_x,\mathbf{e}_y} \hat{P}^-_{\mathbf{j}+\mathbf{e}_y,\mathbf{e}_x} \hat{P}^-_{\mathbf{j},\mathbf{e}_y}$, $\hat{P}^+_{\square,\mathbf{j}} = \hat{P}^-_{\mathbf{j},\mathbf{e}_x} \hat{P}^-_{\mathbf{j}+\mathbf{e}_x,\mathbf{e}_y} \hat{P}^+_{\mathbf{j}+\mathbf{e}_y,\mathbf{e}_x} \hat{P}^+_{\mathbf{j},\mathbf{e}_y}$, and the separation $r$ between the plaquettes is even. For odd $r$, we replace $\hat{P}_{\square,\mathbf{j}+r\mathbf{e}_x} \rightarrow \hat{P}^+_{\square,\mathbf{j}+r\mathbf{e}_x}$.

The correlation functions of the fermion density and the plaquette operator reveal an even more interesting picture than their expectation values. Let us first consider the fermion number correlation function $\mathcal{D}_{\mathbf{n}}(r)$ as shown in Fig. 7 for three different values of $J/t = 0, 2, 6$ and for a range of $m/t$ from $-3$ to $+3$. For $J/t = 0$, we note the absence of any long-range correlation anywhere in the studied range. The longest-ranged correlation exists for $m/t \approx -0.6$, indicating that the peak in the entanglement entropy, which we see in Fig. 5d, is not associated with a phase transition. Turning on a finite value of $J/t$ gives rise to a qualitative difference as illustrated in Fig. 7 (middle) and (right). For $J/t = 2$, there are two distinct regions at $m/t \sim -0.6, 1.0$ that show the longest correlation lengths, which reach more than $30 - 40$ lattice spacings. The region in between, around $m/t \sim 0$, instead has considerably shorter correlation length. We can also see this behavior in the correlation function profiles for the fermion number correlator in Fig. 8 (left), where the decay of the correlation function is orders of magnitude slower at the values of $m/t \sim -0.6$ and $1.0$, and approximates a power law.

This behavior is consistent with our observation, as well as the conjecture made in Ref. [51], about the existence of a narrow phase where a disordered liquid could exist. We substantiate this claim by showing in detail, the behaviour of the various order parameters in Appendix E. To decipher possible symmetry breakings, we compute the structure factors (the Fourier transform of relevant correlation functions) to show that no other lattice or internal symmetry is broken, strengthening the idea that a liquid phase indeed exists there. As $J/t$ increases, the peaks move closer, as we see in Fig. 7 (left and middle panels) and ultimately fuse into a narrow peak shown in Fig. 7 (right). These observations form the basis of the schematic phase diagram sketched in Fig. 2, where we show this phase as an island getting narrower until it disappears.

We can get a similar insight into the phases by looking at the plaquette correlation functions as shown in Fig. 9 for the same values of $J/t$ and $m/t$ as the fermion number correlators. From this figure, it becomes clear that the regimes $J/t = 0$ and $J/t > 0$ are qualitatively different. In the former region, shown in the leftmost panel of Fig. 9, no clear long-range correlation of the gauge fields, either in the QDM-like or the QLM-like phase, can be discerned. Also, as illustrated by the curve corresponding to $J/t = 0$, $m/t = 0$ in Fig. 8 (right), the correlation between the gauge fields decays rapidly with distance, possibly indicating a large gap for this regime.

For finite $J/t$ the gauge fields also exhibit their own distinct behavior. For the case of $J/t > 0$, there is a stronger correlation within the QDM-like phase, indicating that the electron-positron condensed vacuum is capable of causing a stronger correlation among gauge fields at larger physical separations. This behavior is also clear from Fig. 8 (right) where the correlation functions of the plaquettes are plotted for a few values of the coupling. These curves also clearly mark out the zero and the non-zero $J/t$ regimes, and support our claim that the presence of the fermions (whether in the QDM-like phase or the pure gauge QLM-like phase) causes a larger correlation, which falls off orders of magnitude slower. This observation is consistent with the expectation that making the charges mobile in the QDM could lead to the development of a phase where the electrons have collective excitations, which is relevant to building models for high-$T_c$ superconductivity [68, 79, 80]. There is minimal difference between the correlation functions in the QDM-like and the QLM-like phase (the former is slightly stronger), but it is unclear if this difference survives on extending the circumference of the cylinder used in the numerics. Contrary to the fermionic correlators, which show a marked decrease for $m/t \sim 0$, the gauge fields continue to interact strongly in the liquid-like disordered phase. Further studies of scaling of the spectral gaps of various fermion and gauge field operators would be very valuable to fully deciphering the nature of this phase.

## 5 Discussion and Conclusions

In this paper, we have explored the changes in the phases arising due to the coupling of single-flavored fermions to spin-1/2 gauge fields. As we explained before, the fermion mass term enables us to energetically tune the dynamics between the QDM-like and the pure gauge QLM-like physics. We generically find that the inclusion of fermions causes large correlations to build up even in regimes where the fermion mass is very small. This is consistent with previous observations, and lends credence to the expectation that doping the quantum dimer model could give rise to a long-range correlation between the electrons, leading to some form of superconductivity [79, 80]. A particularly interesting aspect of our work is the conjecture of a novel phase for small values of both $J/t$ and $m/t$, where the model remains strongly interacting without breaking any symmetries, and with power law-like correlations in the fermion number and the flippability of the gauge fields. The phase boundary of this phase with the con-

ventional QDM-like and pure gauge QLM-like phases displays a long correlation length raising the possibility that there could be a second-order phase transition between them. This is the scenario we have depicted schematically in Fig. 2. An even more interesting possibility would be if the two phase transition lines would end at a tricritical point. However, a more careful study of different finite sizes as well as the gaps would be necessary to confirm or refute this scenario. We defer this to a future study.

A central motivation for the present study is related to the rapid advances in the quantum simulation of U(1) lattice gauge theories in one spatial dimension, with several pioneering experiments [63,64,81,82] having opened the way also towards observing many-body effects such as quantum phase transitions and quantum thermalization. In parallel, there has been a strong effort towards developing feasible quantum-simulation proposals for U(1) lattice gauge theories in two spatial dimensions [69,83–91], with some first proof-of-principle experimental realizations [92]. In view of this rapid development, it becomes an urgent matter to understand the higher-dimensional phase diagrams of feasible gauge theory models that are likely the first targets of larger-scale experiments. Our study discusses a paradigmatic model, with fermions interacting with quantum links, and provides an opportunity in two spatial dimensions where experimental efforts would be welcome to explore the phase diagram and the real-time dynamics together with the conventional theoretical approaches.

## Acknowledgments

This work is part of and supported by Provincia Autonoma di Trento, the ERC Starting Grant StrEnQTh (project ID 804305), the Google Research Scholar Award ProGauge, and Q@TN — Quantum Science and Technology in Trento. T. H. was supported by AFOSR grant number FA9550-18-1-0064. T. H. would like to thank Ian McCulloch for useful discussions. The data for this manuscript is available in open access at Ref. [93].

## A  Winding Number

In this section, we study the expectation values of the winding-number operators. Strictly speaking, the operators as defined in Eq. (10) only make sense for a pure gauge theory. However, we define slightly modified quantities, which we expect to pick up the condensation of electric fluxes:

$$W_x = \frac{1}{L_x L_y} \sum_{y_0} \left| \sum_{\mathbf{j}=(x,y_0)} \langle \hat{E}_{\mathbf{j},\mathbf{e}_y} \rangle \right|, \tag{19a}$$

$$W_y = \frac{1}{L_x L_y} \sum_{x_0} \left| \sum_{\mathbf{j}=(x_0,y)} \langle \hat{E}_{\mathbf{j},\mathbf{e}_x} \rangle \right|. \tag{19b}$$

The sum of the electric fluxes in the above two definitions are taken along the $x$-axis at a fixed value of $y = y_0$, and along the $y$-axis at a fixed value of $x = x_0$. This picks up a net flux along a vertical (horizontal) cut bisecting the horizontal (vertical) links. Finally, we sum them along all the cuts, and divide by the volume. In the iMPS calculation, the winding numbers as defined in Eq. (19) are observed to have a value of 0 within the numerical precision of our simulations for all values of $J$ and $m$ that are simulated in the paper.

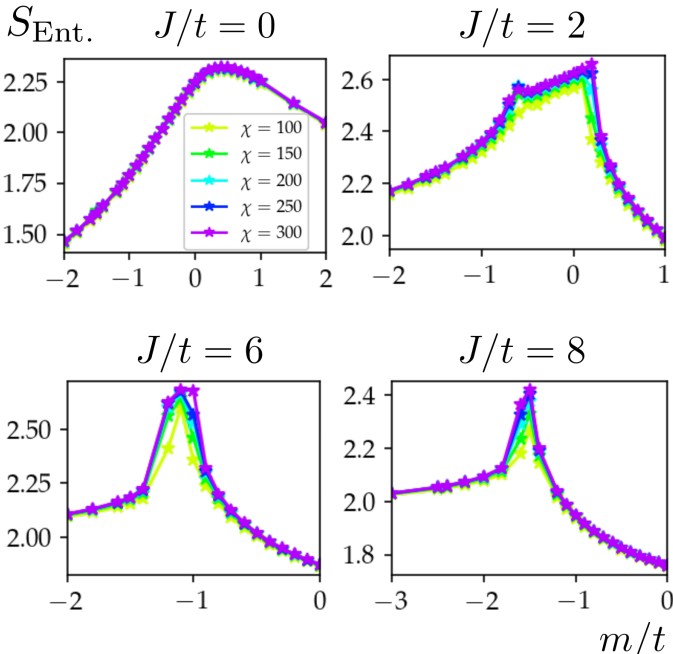

Figure 10: Entanglement entropy near $m = 0$ for bond dimensions $\chi = 100, 150, 200, 250,$ and $300$ for $J/t = 0, 2, 6,$ and $8$. For $J/t = 0$, no strong peak in the entanglement entropy is observed, suggesting that the quantum link model smoothly crosses over to the quantum dimer model without criticality. For finite $J/t$, on the other hand, a trace of criticality is expected with the increasing entanglement entropy with respect to the bond dimension at the points where $\langle \hat{\mathcal{O}}_F \rangle = 0$. The peak in the entropy persists for the region in the phase diagram where the sign of $\langle \hat{\mathcal{O}}_F \rangle$ inverts from the quantum dimer limit.

## B Convergence with the bond dimensions

The calculations in the main text are done with the bond dimension $\chi = 300$. In Fig. 10, we show how the entanglement entropy scales with the bond dimension of the MPS near $m = 0$ at $J/t = 0, 2, 6,$ and $8$. At $J/t = 0$, the entanglement entropy converges with the bond dimension at all considered values of $m/t$, indicating that the quantum dimer limit smoothly crosses over into the quantum link limit, with no observable criticality. At finite $J/t$, in contrast, there exists a region where the entanglement entropy peaks, and where no convergence is observed within the values of $\chi$ considered. This region narrows as $J/t$ increases, until the two points characterized by the inversions in the sign of $\langle \hat{\mathcal{O}}_F \rangle$ in Fig. 5 merge and disappear.

## C Effect of cylinder circumference $L_y$

### C.1 $L_y = 2$ geometry

As shown in Fig. 11, the model with a cylinder with $L_y = 2$ possesses a similar phase diagram as the model with $L_y = 4$. Both chiral condensate and $\langle \hat{\mathcal{O}}_F \rangle$ vanishes near $m/t = 0$, which further emphasizes that broken chiral symmetry is largely controlled by the staggered mass term (Fig. 11a and b). For $\langle \hat{\mathcal{O}}_F \rangle$, however, the region where its sign inversion occurs is not associated with a region with large entanglement entropy for the model with $L_y = 2$ (Fig. 11c and d).

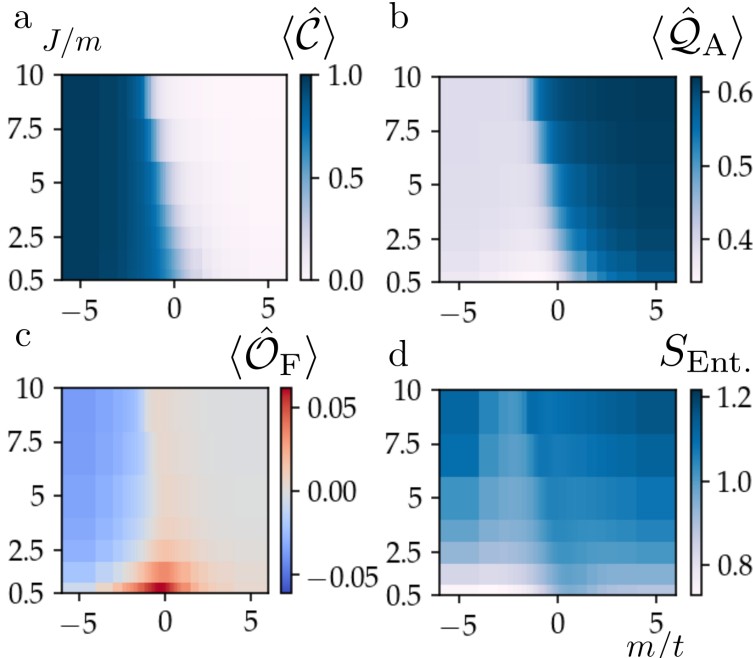

Figure 11: Order parameters of the model with $L_y = 2$. a. Chiral condensate $\langle \hat{\mathcal{C}} \rangle$. b. $\langle \hat{\mathcal{O}}_F \rangle$ detects the flippable plaquettes ($\langle \hat{\mathcal{O}}_A \rangle = 0$) c. $\langle \hat{\mathcal{Q}}_A \rangle$ are sensitive to the (anti) clockwise ordering of the plaquettes, while $\langle \hat{\mathcal{Q}}_F \rangle = 0$, and takes a positive value, when even (odd) site favors anticlockwise (clockwise) orientation. d. Entanglement entropy of a bipartition along the $y$-direction. Unlike in the case of a 4-leg cylinder, the island near $m = 0$ with inverted sign of $\langle \hat{\mathcal{O}}_F \rangle$ is not accompanied by large bipartite entanglement entropy. The region increases its presence on higher $J/t$ as the number of sites along the circumference doubles from $L_y = 2$ to $L_y = 4$.

## C.2 Emergent phase on $L_y = 4$ geometry

In the main text, $\langle \hat{\mathcal{O}}_A \rangle$ is not explored because its value is 0 within the numerical precision of our simulations at $J = 0$. The operator $\hat{O}_A$ is an order parameter that detects the broken shift (chiral) symmetry of the plaquettes in terms of how flippable they are. In Fig. 12a, we show that such a symmetry breaking occurs for $L_y = 4$, while $\langle \hat{\mathcal{O}}_A \rangle = 0$ for $L_y = 2$. For $L_y = 2$, both quantities vary smoothly. For $L_y = 4$ on the other hand, there is what appears to be a first-order phase transition with the symmetry breaking characterized by $\langle \hat{\mathcal{O}}_A \rangle$ (Fig. 12b). The peak in the entanglement entropy near $m = 0$ at and around the island of the sign inversion of $\langle \hat{\mathcal{O}}_F \rangle$, which occurs for $L_y = 4$ but not $L_y = 2$, is a signature of that these phases are emergent in the two-dimensional lattice geometry.

Furthermore, the profile of the electric field and fermion number in Fig. 12c further emphasizes the qualitative difference between $L_y = 2$ and $L_y = 4$. The columnar and uniformly distributed flippable plaquettes cannot be distinguished with $\langle \hat{\mathcal{O}}_A \rangle$, which are equivalently 0 for $L_y = 2$ and $L_y = 4$ at $m = -4$ (i,iv) and $m = 0$ (ii,v). The difference between (iv) and (v) is captured by the order parameter $\langle \mathcal{Q}_F \rangle$ as its sign inversion. For the large values of $m/t$, the shift symmetry explicitly breaks for $L_y = 4$ (vi), while the values of $\langle \hat{\mathcal{O}}_j \rangle$ are uniform across the plaquettes for $L_y = 2$.

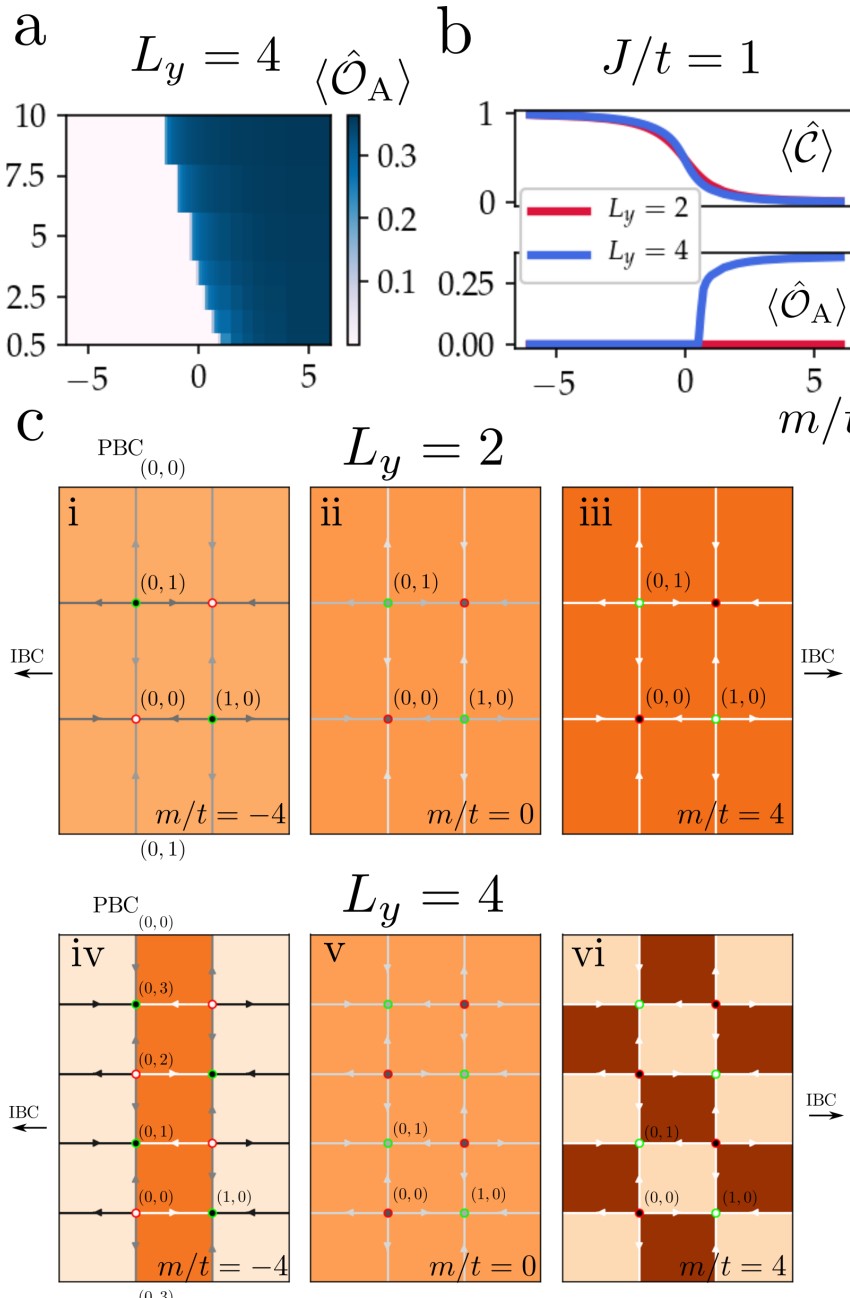

Figure 12: The emergent phases at $L_y = 4$, which are not present for $L_y = 2$. a. $\langle \hat{\mathcal{O}}_A \rangle$, which detects the flippable plaquettes. b. Chiral Condensate $\mathcal{C}$ (upper) and $\langle \hat{\mathcal{O}}_A \rangle$ (lower) at $J/t = 1$ for $L_y = 2$ (red) and $L_y = 4$ (blue). The first-order phase transition is captured by the order parameter $\langle \hat{\mathcal{O}}_A \rangle$, while the chiral condensate approaches 0 smoothly. c. Electric flux and fermion number profile for $J/t = 1$ for $L_y = 2$ (i,ii,iii) and $L_y = 4$ (iv,v,vi). For $L_y = 4$, the columnar phase (iv) melts near $m/t = 0$ as fermionic chiral symmetry vanishes (v). The QLM-like phase recovers as $m/t$ increases (vi). Such rich phase transitions do not occur in the process of increasing $m$ for $L_y = 2$ (i,ii,iii). The plaquettes between $\mathbf{j}_y = 0$ and $\mathbf{j}_y = L_y - 1$ are duplicated at the top and the bottom to clearly illustrate the connections in the periodic boundary condition imposed on the $y$-direction.

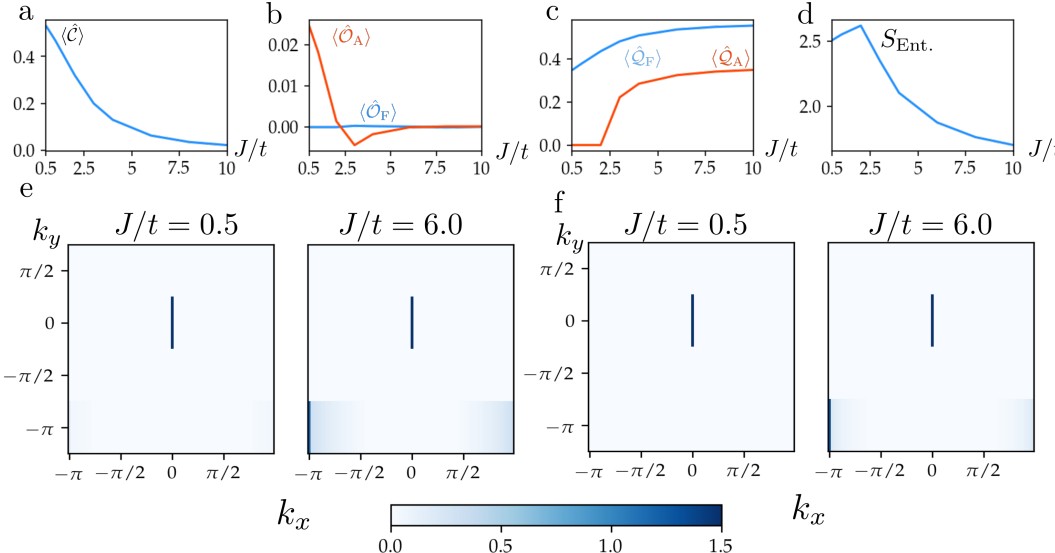

Figure 13: The behavior of the order parameters and structure factors for $L_y = 4$ along the $m/t = 0$ cross section. a. Chiral Condensate $\langle \hat{\mathcal{C}} \rangle$, b. the plaquette operators $\langle \hat{\mathcal{O}}_F \rangle$ and $\langle \hat{\mathcal{O}}_A \rangle$, c. the plaquette operators $\langle \hat{\mathcal{Q}}_F \rangle$ and $\langle \hat{\mathcal{Q}}_A \rangle$, and d. bipartite entanglement entropy of the cut along the $y$-direction. While the chiral condensate approaches 0 smoothly, the plaquette order $\langle Q_A \rangle$ rises sharply at $J/t \approx 2$. This sharp raise is accompanied by a peak in the entanglement entropy, and it monotonically decreases $J/t > 2$. Similarly, the clockwise plaquette order $\mathcal{O}_A$ changes sign at the point $J/t \approx 2$, and approaches to 0 from the negative direction as $J/t$ increases. e. The magnitude of the structure factor $\log_{10} S^{(P_\square)}(k_x, k_y)$ and f. the magnitude of the structure factor $\log_{10} S^{(U_\square)}(k_x, k_y)$, for the cell of size $L_x = 80$ by $L_y = 4$. The dominant $\mathbf{k} = (0, 0)$ mode in $J/t = 0.5$ and emerging $\mathbf{k} = (-\pi, -\pi)$ order in $J/t = 6$ in both e. and f. further support the recovery of symmetry in the region near $m = 0$ for the small values of $0 < J/t < 2$.

## D    Comparison with exact diagonalization

In this section, we report on benchmarking the iMPS code with ED results. As remarked before, when one considers the fermionic Hilbert space together with the gauge fields, it becomes very difficult to perform ED for sufficiently large lattice sizes. Therefore, we performed ED in the limit of pure QDM ($m/t \to -\infty$) and pure QLM ($m/t \to +\infty$) where it is considerably more manageable. For a numerical comparison, we cross-checked the ED results to their iMPS counterparts for large mass regimes, and accounted for the deviation using $1/m$ corrections.

Consider the case with $J/t = 0$, where the ED for pure gauge theory just involves taking a naive average over all basis states diagonal in the electric flux basis. In the QDM limit, we get $\langle \hat{\mathcal{O}}_F \rangle = 0.3030$ from the ED results, while the iMPS value for $m/t = -6$ is $\langle \hat{\mathcal{O}}_F \rangle = 0.2925$. To make the comparison, we have considered the ground state in both the ED and the iMPS results. In the QLM limit on the other hand, for $J/t = 0$, ED gives $\langle \hat{\mathcal{O}}_F \rangle = 0.4606$ while iMPS for $m/t = 6$ yields $\langle \hat{\mathcal{O}}_F \rangle = 0.5191$. The deviation in the case of QDM is about 3%, which is comparable to $\mathcal{O}(1/m^2)$, where $m$ is the bare fermion mass. The deviation in the case of QLM is larger, at about 12%, indicating that the fluctuations due to the fermion mass are still important in this regime.

For $J/t > 0$, the physics is different, as we already inferred from the expectation values. Typically, there is very little dependence (less than 1% for both the QDM and the QLM cases) on

the exact value of $J/t$ in the results of the cylinder with $L_y = 4$. For the QDM-like phase with $m/t = -6$, the expectation values at $J/t = 2.0, 4.0, 6.0$ are $\langle \hat{\mathcal{O}}_F \rangle = 0.3452, 0.3455, 0.3457$. These values compare very favorably with the ED results (i.e., for $m/t = -\infty$) where $\langle \hat{\mathcal{O}}_F \rangle = 0.3544$.

For the QLM-like phase with $m/t = 6$, the expectation values at $J/t = 2.0, 4.0, 6.0$ are $\langle \hat{\mathcal{O}}_F \rangle = 0.5617, 0.5628, 0.5635$. Once again, these values agree well with the ED results (i.e., for $m/t = \infty$) where $\langle \hat{\mathcal{O}}_F \rangle = 0.5776$. For finite $J/t$ the deviation of ED results from the iMPS results is at the level of 3% for both the QDM and the QLM-like phases, with a negative sign of the deviation. We find these levels of agreement satisfactory, particularly in view of the lattice sizes considered.

# E  Order Parameters and Structure factors at vanishing bare mass

In this section, we discuss the behaviors of the order parameters in the regime where the bare mass vanishes, $m/t = 0$. As we remark in the main text, we observe an anomalous behavior in all physical quantities in this regime for small values of $J/t > 0$, accompanied by large bipartite entanglement entropy. We hypothesize that this region corresponds to a liquid-like phase. To convince the reader further that this is indeed the case, we present the order parameter along the line $m/t = 0$ for the cylinder with $L_y = 4$.

In Fig. 13 a. to d., we show the three order parameters and the bipartite entanglement entropy discussed in the main text: Chiral condensate $\langle \hat{C} \rangle$, $\langle \hat{\mathcal{O}}_{F/A} \rangle$, $\langle \hat{\mathcal{Q}}_{F/A} \rangle$, and $S_{\text{Ent.}}$. In this regime, the occupation number of the fermions is closely correlated with the electric flux of the gauge field. For $J/t = 0$, the magnetic field term does not play a role, and the fermionic hops, whenever allowed by the gauge fields, decide the features of the strongly correlated emergent liquid phase. For small but finite $J/t$, the plaquette flipping magnetic field terms are also present, adding to the gauge field fluctuations. Due to the larger allowed Hilbert space of the gauge links in the QLM limit, the configurations of the fermions prefer the QLM-like configurations. This moves the transition point of the chiral condensate towards large and negative $m/t$ values (Fig. 13 a.). The clock- and anticlockwise plaquette orders, on the other hand, do not show a strong antiferromagnetic order for the small values of $J/t$ (Fig. 13 b. and c.). The symmetry breaking in the plaquette occurs near $J/t = 2$, and this accompanies the decay of bipartite entanglement entropy.

Finally, we show the result of the structure factor $S(k_x, k_y)$. We define two different structure factors, following from the two point correlation functions of the local operators, $\hat{P}_{\square,\mathbf{j}}$ and $\hat{U}_{\square,\mathbf{j}}$ defined in the main text:

$$S^{(P_\square)}(k_x, k_y) = \sum_{\mathbf{j},\mathbf{j}'} \langle \hat{P}_{\square,\mathbf{j}} \hat{P}_{\square,\mathbf{j}'} \rangle \exp\left(\mathbf{k} \cdot (\mathbf{j} - \mathbf{j}')\right), \tag{20}$$

$$S^{(U_\square)}(k_x, k_y) = \sum_{\mathbf{j},\mathbf{j}'} \langle \hat{U}_{\square,\mathbf{j}} \hat{U}^\dagger_{\square,\mathbf{j}'} \rangle \exp\left(\mathbf{k} \cdot (\mathbf{j} - \mathbf{j}')\right). \tag{21}$$

Shown in Fig. 13 e. and f. are the structure factors Eq. (20) and Eq. (21) plotted for $J/t = 0.5$ and $J/t = 6$. An order at $(-\pi, -\pi)$, which is present at $J/t = 6$, diminishes as the value of $J/t$ decreases towards $J/t = 0$. This further strengthens our hypothesis on the existence of a liquid-like phase near the $m/t = J/t = 0$ point.

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
