# Peer review of "Ground-state phase diagram of quantum link electrodynamics in $(2+1)$-d"

_SciPost Physics, doi:SciPost Phys. 13, 017 (2022)_

## Round 1 · Referee Report · Anonymous · 2022-1-18

Strengths

See the report section

Weaknesses

See the report section

Report

In the manuscript, the Authors simulate numerically a two-dimensional spin-½ quantum link lattice QED, on a cylinder, with infinite-DMRG. They perform an in-depth phase analysis, using meaningful quantitative phase detectors, and uncover a rich phase diagram, including some potentially-exotic phases. The work is rigorous, scientifically sound, deals with a currently hot topic, and is fairly contextualized.
However, before I can recommend publication, the Authors should fix a few drawbacks in the presentation that hinder the delivery of the contents to the reader. Most of my concerns are about the way physics (of the model, of the results) is presented, so they can hardly be ignored. I summarize my biggest worries in the following list:

(I) The QLM Hamiltonian at equation (4) presents some issues with the typical way the lattice QED is presented. In particular, the matter hopping terms do not exhibit the typical phases that one would expect from the Dirac equation, or more precisely Susskind's lattice version (Phys. Rev. D 16, 3031 (1976), equation (3.5)). In Susskind's picture, if one direction, say x, the hoppings are imaginary and straight, in the other direction, say y, they should be real and staggered.
Is the authors’ picture equivalent? Could the authors please elaborate on this choice?

(II) I would like the author to provide more context and details concerning their choice of finite-spin representation for the gauge field. Yes, I agree that spin-1/2 is the simplest non-trivial degree of freedom somewhat allowing a representation of the gauge algebra (albeit losing the electric energy density as a relevant operator), but there is definitely more to it. My intuition is that integer-spin representations (e.g. ref. 54) and half-integer spin representation may exhibit different phases at the thermodynamically limit, which should vanish ONLY in the Wilson-continuum limit. I think the authors should expand on this important point.

(III) Still about the spin-½ representation, the authors state that “the electric field energy term is a constant and can be ignored”. I wish it were that simple. The way I see it, the truncation of the gauge field to a spin-½ is an energy-density cutoff, and we are saying that the energyscale to the next two electric field states (the new ones that would appear in the spin-3/2 reoresentation) is the largest local energyscale of the system, and THUS can be ignored. This clearly imposes conditions on the elecrtic energy coupling. The authors write g^2/2 but according to Kogut-Susskind (Ref [55] Equations (6.6) and (7.1)) it can be written as g^2/2a including the lattice spacing a. At the same time, the magnetic energy coupling (the authors simply call it J) is RELATED to this coupling because it should read 4a/g^2. Therefore, for the spin-½ representation to be meaningful in the sense of QED, some constraints on g^2 and on the lattice spacing ‘a’ must be included. The authors should definitely state them.

(IV) The Shift symmetry, as it is written in equation (7) does not seem to be correct to me. The bare mass term is not invariant under either of those transformations. Additionally, Susskind’s lattice fermi field is meant to have four sublattices (arranged like a square), each corresponding to a member of the Dirac 4-spinor. Therefore a shift symmetry of one site SHOULD NOT exist (while 2-site in x and/or 2-site in y should). The authors must elaborate on this point.

(V) I can not find any mention of the global excess charge, either as a symmetry (which exists) or as a quantum number to be taken care of. The way I see it, if the simulations were finite-length cylinders (that could be done with finite-size DMRG), the authors could control `some’ degree of excess charge, either positive or negative, by setting appropriate conditions on the EM quantum fields at the two edges of the cylinder (aka Von Neumann boundary conditions). However, the authors are tackling the problem for an infinite cylinder (via infinite-DMRG), which, if I understand correctly, automatically leads them to the sector of zero excess charge, the only sector where the gauge field can be really translationally invariant. Is this true? If it is, I believe the authors should include the latter observation in the manuscript.

(VI) Figure 2 is a nice phase diagram, but it is plotted as a function of the effective couplings (T,M,J) . IF the authors care about visibility to the high-energy community, they may consider exhibiting the same plot a second time, but with the bare couplings of the lattice QED theory (QED dimensionless coupling g, lattice spacing a, bare mass m).
Conversion rates are: T => c \hbar /( 2a ) ; M => m c^2 ; J = 4 c \hbar /( a g^2 ).
You are welcome.

(VII) The chiral condensate: Equation (15) would be much easier to read if it would be explained that the whole expression within the sum is simply the modulus of the electrical charge at site j.

(VIII) The flippability operators O_j and Q_j seem to be very interesting, and capable of detecting interesting thermodynamical phases. However, from equation (17) it seems that you are only interested in the Brillouin zone points (0,0) and (\pi,\pi). Since the authors have calculated the whole 2-point correlations functions in real space, they should consider exhibiting the complete static structure factor S(k_x, k_y) \sim sum_{j, j’} < O_j O_(j+j’) > e^{i k \cdot j’}. This way, they should be able to immediately identify if another wavevector is relevant (beside the Brillouin center and corner). Was this check made?
ESPECIALLY figure 6 (left panel) seems to point towards an order at the (\pi, 0) point.

(IX) Figure 6. I am understanding that Ly = 4 and PBC on y-direction, correct? So, in this case are the top plaquettes and the bottom plaquettes, in figure 6, actually the same plaquettes? If yes, the picture is somewhat unclear. Authors should consider re-editing.

(X) Finally, it could be helpful to have some cartoon picture to intuitively explain, with the help of visual drawing, the prominent feature of the various phases detected by the authors. Figures 8 and 9 attempt to do so, but being actual data plots, they convey the message poorly. Authors should consider sketching an actual drawing.

Requested changes

See the report

  • validity: good
  • significance: good
  • originality: good
  • clarity: good
  • formatting: good
  • grammar: good

Author:  Tomohiro Hashizume  on 2022-05-28  [id 2533]

(in reply to Report 1 on 2022-01-18)

>> In the manuscript, the Authors simulate numerically a two-dimensional spin-½ quantum
>> link lattice QED, on a cylinder, with infinite-DMRG. They perform an in-depth phase
>> analysis, using meaningful quantitative phase detectors, and uncover a rich phase
>> diagram, including some potentially-exotic phases. The work is rigorous, scientifically
>> sound, deals with a currently hot topic, and is fairly contextualized.
>> However, before I can recommend publication, the Authors should fix a few drawbacks in
>> the presentation that hinder the delivery of the contents to the reader. Most of my
>> concerns are about the way physics (of the model, of the results) is presented, so they
>> can hardly be ignored. I summarize my biggest worries in the following list:

We would like to thank the Referee for recognizing the relevance of our work to the international research effort in this highly active field. We appreciate the shortcomings pointed out, and provide our responses below, as well as a list of changes made to address the issues pointed out.

>> (I) The QLM Hamiltonian at equation (4) presents some issues with the typical way the
>> lattice QED is presented. In particular, the matter hopping terms do not exhibit the
>> typical phases that one would expect from the Dirac equation, or more precisely Susskind's
>> lattice version (Phys. Rev. D 16, 3031 (1976), equation (3.5)). In Susskind's picture,
>> if one direction, say x, the hoppings are imaginary and straight, in the other direction,
>> say y, they should be real and staggered.
>> Is the authors’ picture equivalent? Could the authors please elaborate on this choice?

Thanks to the Referee for ensuring our presentation of lattice QED is standardized. In fact, our equation (4) is a more symmetric form equation (3.5) of Susskind's paper Phys. Rev. D 16, 3031 (1976). If one follows the subsequent equations Susskind's paper, after a series of relabellings and Fourier transforms one obtains Eq. (3.17) of the same paper, which is the equation of motion for the Hamiltonian of our equation (4) without the gauge fields. By using equation (4), we have chosen to follow a way of representing lattice QED that is common to certain standard textbooks for the lattice-gauge theory community, see for example, equation (6.47) of "Lattice Methods for Quantum Chromodynamics" by DeTar and DeGrand (World Scientific).

>> (II) I would like the author to provide more context and details concerning their choice
>> of finite-spin representation for the gauge field. Yes, I agree that spin-1/2 is the simplest
>> non-trivial degree of freedom somewhat allowing a representation of the gauge algebra
>> (albeit losing the electric energy density as a relevant operator), but there is definitely
>> more to it. My intuition is that integer-spin representations (e.g. ref. 54) and half-integer
>> spin representation may exhibit different phases at the thermodynamic limit, which should
>> vanish ONLY in the Wilson-continuum limit. I think the authors should expand on this important point.

Indeed, we agree with the Referee that integer and half-integer spins exhibit different physics, sometimes even in the thermodynamic limit. In fact, as demonstrated in arXiv:2104.00025, the spin-1/2 representation can be interpreted as a non-trivial topological angle $\theta=\pi$.

In using the spin-1/2 representation, we follow the quantum link formulation of gauge theories (introduced in its current form in S. Chandrasekharan and U.-J. Wiese, Nuclear Physics B, 17 (1997)), where it is shown that a valid lattice gauge theory with U(1) gauge symmetry is obtained even in when using the simplest choice of S=1/2. One of our aims was precisely to identify the type of physics that is obtained by this choice.

We have added a discussion on this aspect in the beginning of Section II. Further remarks on how one can reach the Wilson limit are explained in the text below Eq. (3).

>> (III) Still about the spin-½ representation, the authors state that “the electric field energy
>> term is a constant and can be ignored”. I wish it were that simple. The way I see it, the truncation
>> of the gauge field to a spin-½ is an energy-density cutoff, and we are saying that the energyscale
>> to the next two electric field states (the new ones that would appear in the spin-3/2 reoresentation)
>> is the largest local energyscale of the system, and THUS can be ignored. This clearly imposes
>> conditions on the elecrtic energy coupling. The authors write g^2/2 but according to Kogut-Susskind
>> (Ref [55] Equations (6.6) and (7.1)) it can be written as g^2/2a including the lattice spacing a.
>> At the same time, the magnetic energy coupling (the authors simply call it J) is RELATED to this
>> coupling because it should read 4a/g^2. Therefore, for the spin-½ representation to be meaningful
>> in the sense of QED, some constraints on g^2 and on the lattice spacing ‘a’ must be included. The
>> authors should definitely state them.

We thank the Referee for bringing this point up, which is somewhat subtle and not always fully appreciated. Firstly, we would like to emphasize that the quantum link models generalize the Wilson-Kogut-Susskind formulation by introducing a gauge invariant parameter, which is the representation of the quantum links. For Abelian gauge theories, this is the spin-representation S, such that each value of S defines a completely independent theory. The spin representation S=1/2 and S=1, for example, defines two independent microscopic theories. If one lives in a "universe" with the spin-1/2 model, then the electric flux can only take the values E = +1/2 and E = -1/2, since a spin-1/2 gauge link cannot be raised higher. On the other hand, for the S=1 model, g^2 coupling would have contributed non-trivially since there would have been three states of the electric flux E = -1, 0, +1. These can also be viewed as independent theories parameterized by S, where the choice of the representation decides the allowed values. Of course, if one is interested in achieving the traditional continuum limit, then one should rescale the couplings appropriately, as is done in arXiv:2104.00025, a related work with some present authors.

Another way of imagining the situation is to recognize that with our microscopic model, we are sitting on the RG trajectory with g^2 = 0, and can only explore the physics of differentfixed points along this axis. This is a natural way of fine-tuning the problem such that one cannot move away from this axis.

Regarding the second comment of the Referee, we would like to point out that the electric and the magnetic field couplings are only related if one flows to a relativistic fixed point. While this is true for the Wilson-Kogut-Susskind theory in taking the 'usual' continuum limit, it need not always be the case. In particular, if the system flows to a non-relativistic fixed point, the magnetic and the electric couplings would typically not be related to each other as in the relativistic case. Therefore, one should not demand the relation of the couplings beforehand, a point not always appreciated in the literature. The inclusion of the lattice spacing is therefore only useful if one flows to the usual relativistic fixed point, where the electric and the magnetic field scale with the related anomalous dimensions. Since we do not assume that we flow to such a fixed point, we have deliberately not included the lattice spacing in the couplings.

To summarize, in this work, we aim to explore the physics away from the traditional continuum limit and therefore do not impose any such restrictions beforehand, or demand the restoration of Lorentz invariance. In particular, this is done keeping in mind non-relativistic fixed points where one can flow to. We explain our reasoning in the beginning of Section II.

>> (IV) The Shift symmetry, as it is written in equation (7) does not seem to be correct to me.
>> The bare mass term is not invariant under either of those transformations. Additionally,
>> Susskind’s lattice fermi field is meant to have four sublattices (arranged like a square),
>> each corresponding to a member of the Dirac 4-spinor. Therefore a shift symmetry of one site
>> SHOULD NOT exist (while 2-site in x and/or 2-site in y should). The authors must elaborate on this point.

The shift symmetry is the analog of the continuous chiral symmetry of a Dirac fermion for the single component staggered fermion. Just as a mass term explicitly breaks the chiral symmetry of a Dirac fermion, so does the mass term break the shift symmetry of the staggered fermion.

As the Referee correctly points out, a four-component Dirac fermion can be formed using four single component staggered fermions. In fact, one can show that the single component staggered fermions (also called Kogut-Susskind fermions in lattice gauge theory community) preserves a subgroup of the full chiral symmetry of the full Dirac fermions, and is an internal symmetry in this sense. The two-site lattice translations (in either x or in y direction) is the usual lattice translation symmetry as also explained in the Manuscript, below the equations 7a,b,c. We have added a few sentences explaining these issues before equation 7, where the shift symmetry is introduced.

>> (V) I can not find any mention of the global excess charge, either as a symmetry (which exists)
>> or as a quantum number to be taken care of. The way I see it, if the simulations were finite-length
>> cylinders (that could be done with finite-size DMRG), the authors could control `some’ degree of
>> excess charge, either positive or negative, by setting appropriate conditions on the EM quantum
>> fields at the two edges of the cylinder (aka Von Neumann boundary conditions). However, the authors
>> are tackling the problem for an infinite cylinder (via infinite-DMRG), which, if I understand correctly,
>> automatically leads them to the sector of zero excess charge, the only sector where the gauge field
>> can be really translationally invariant. Is this true? If it is, I believe the authors should include
>> the latter observation in the manuscript.

As the Referee correctly states, we work in the sector of zero excess charge. This is a common choice in particle physics scenarios, and is explicitly stated in the parageaph below Eq.(6).

>> (VI) Figure 2 is a nice phase diagram, but it is plotted as a function of the effective couplings (T,M,J) .
>> IF the authors care about visibility to the high-energy community, they may consider exhibiting the same
>> plot a second time, but with the bare couplings of the lattice QED theory (QED dimensionless coupling g,
>> lattice spacing a, bare mass m). Conversion rates are: T => c \hbar /( 2a ) ; M => m c^2 ; J = 4 c \hbar /( a g^2 ).
>> You are welcome.

We thank the Referee very much for this suggestion, but---considering the discussion above---we consider it might be somewhat misleading to plot it in terms of effective couplings. Firstly, t is the hopping strength and not the temperature, and moreover the conversions assume the usual relativistic relation between J and g^2, which need not be true. Such a depiction, in our opinion, would only make sense after the fixed points (especially a second order phase transition) has been identified, and once the scaling dimensions have been determined appropriately (to understand especially if it is a relativistic or a non-relativistic fixed point), one can relate the couplings as proposed. Thus, we prefer to retain only the version with the effective couplings (t,M,J). However, we do mention the relativistic scaling in the text and also our motivation of not using the conversion factors, so that it is clear to the reader.

>> (VII) The chiral condensate: Equation (15) would be much easier to read if it would be explained that
>> the whole expression within the sum is simply the modulus of the electrical charge at site j.

We thank the Referee for pointing this out, this has been explained now.

>> (VIII) The flippability operators O_j and Q_j seem to be very interesting, and capable of detecting interesting
>> thermodynamical phases. However, from equation (17) it seems that you are only interested in the Brillouin zone
>> points (0,0) and (\pi,\pi). Since the authors have calculated the whole 2-point correlations functions in real
>> space, they should consider exhibiting the complete static structure factor S(k_x, k_y) \sim
>> sum_{j, j’} < O_j O_(j+j’) > e^{i k \cdot j’}. This way, they should be able to immediately identify if another
>> wavevector is relevant (beside the Brillouin center and corner). Was this check made?
>> ESPECIALLY figure 6 (left panel) seems to point towards an order at the (\pi, 0) point.

The Referee raises a very good point, and we now show the results for the structure factor computed from both the $O_j$ (eq. 16a) and the $U_\square$ (eq. 5) operators. This is particularly interesting to show for the proposed spin-liquid phase, which then displays the only ordering at $(0,0)$, showing that no other symmetry is broken. We have a special appendix E, in which these results are shown in detail, together with the order parameters in the proposed spin liquid phase. This also answers a query of the second referee. This Appendix is also refereed to in the main text, to indicate where to look for the details without breaking the flow.

Moreover, Fig 6 (left panel) it should be noted that the links, represented by the arrows, point to the top and bottom (for vertical links) and left and right (for the horizontal links) as we move one unit in the y-direction indicating that the order is more (\pi,\pi) instead of (0,0). Similarly the fermion occupation also alternate between 0 and 1 indicating to the same order. Noting the shift operator introduced earlier, it is easy to conclude that the discrete chiral symmetry is broken. This is also visible in the structure factors.

>> (IX) Figure 6. I am understanding that Ly = 4 and PBC on y-direction, correct? So, in this case are the top
>> plaquettes and the bottom plaquettes, in figure 6, actually the same plaquettes? If yes, the picture is somewhat
>> unclear. Authors should consider re-editing.

We thank the Referee for pointing out the problem with the earlier figure, which we had missed. We have now remade this figure and the co-ordinates of the various points are written out to explain the periodicity in the y-direction.

>> (X) Finally, it could be helpful to have some cartoon picture to intuitively explain, with the help of visual
>> drawing, the prominent feature of the various phases detected by the authors. Figures 8 and 9 attempt to do so,
>> but being actual data plots, they convey the message poorly. Authors should consider sketching an actual drawing.

Thanks to the Referee for pointing this out. We include some cartoon sketches to elaborate these points.

We hope that the Referee is satisfied with our responses to their comments, and would consider recommending publication of our paper in SciPost Physics.

Author:  Tomohiro Hashizume  on 2022-05-20  [id 2498]

(in reply to Report 1 on 2022-01-18)
Category:
remark
answer to question

>> In the manuscript, the Authors simulate numerically a two-dimensional spin-½ quantum
>> link lattice QED, on a cylinder, with infinite-DMRG. They perform an in-depth phase
>> analysis, using meaningful quantitative phase detectors, and uncover a rich phase
>> diagram, including some potentially-exotic phases. The work is rigorous, scientifically
>> sound, deals with a currently hot topic, and is fairly contextualized.
>> However, before I can recommend publication, the Authors should fix a few drawbacks in
>> the presentation that hinder the delivery of the contents to the reader. Most of my
>> concerns are about the way physics (of the model, of the results) is presented, so they
>> can hardly be ignored. I summarize my biggest worries in the following list:

We would like to thank the Referee for recognizing the relevance of our work to the
international research effort in this highly active field. We appreciate the shortcomings
pointed out, and provide our responses below, as well as a list of changes made to address
the issues pointed out.

>> (I) The QLM Hamiltonian at equation (4) presents some issues with the typical way the
>> lattice QED is presented. In particular, the matter hopping terms do not exhibit the
>> typical phases that one would expect from the Dirac equation, or more precisely Susskind's
>> lattice version (Phys. Rev. D 16, 3031 (1976), equation (3.5)). In Susskind's picture,
>> if one direction, say x, the hoppings are imaginary and straight, in the other direction,
>> say y, they should be real and staggered.
>> Is the authors’ picture equivalent? Could the authors please elaborate on this choice?

Thanks to the Referee for ensuring our presentation of lattice QED is standardized.
In fact, our equation (4) is a more symmetric form equation (3.5) of Susskind's paper
Phys. Rev. D 16, 3031 (1976). If one follows the subsequent equations Susskind's paper,
after a series of relabellings and Fourier transforms one obtains Eq. (3.17) of the same paper,
which is the equation of motion for the Hamiltonian of our equation (4) without the gauge fields.
By using equation (4), we have chosen to follow a way of representing lattice QED that is common
to certain standard textbooks for the lattice-gauge theory community, see for example, equation (6.47)
of "Lattice Methods for Quantum Chromodynamics" by DeTar and DeGrand (World Scientific).

>> (II) I would like the author to provide more context and details concerning their choice
>> of finite-spin representation for the gauge field. Yes, I agree that spin-1/2 is the simplest
>> non-trivial degree of freedom somewhat allowing a representation of the gauge algebra
>> (albeit losing the electric energy density as a relevant operator), but there is definitely
>> more to it. My intuition is that integer-spin representations (e.g. ref. 54) and half-integer
>> spin representation may exhibit different phases at the thermodynamic limit, which should
>> vanish ONLY in the Wilson-continuum limit. I think the authors should expand on this important point.

Indeed, we agree with the Referee that integer and half-integer spins exhibit
different physics, sometimes even in the thermodynamic limit. In fact, as demonstrated in
arXiv:2104.00025, the spin-1/2 representation can be interpreted as a non-trivial topological
angle $\theta=\pi$.
In using the spin-1/2 representation, we follow the quantum link formulation of gauge
theories (introduced in its current form in S. Chandrasekharan and U.-J. Wiese, Nuclear Physics
B, 17 (1997)), where it is shown that a valid lattice gauge theory with U(1) gauge symmetry is
obtained even in when using the simplest choice of S=1/2. One of our aims was precisely to identify
the type of physics that is obtained by this choice.

We have added a discussion on this aspect in the beginning of Section II. Further remarks
on how one can reach the Wilson limit are explained in the text below Eq. (3).

>> (III) Still about the spin-½ representation, the authors state that “the electric field energy
>> term is a constant and can be ignored”. I wish it were that simple. The way I see it, the truncation
>> of the gauge field to a spin-½ is an energy-density cutoff, and we are saying that the energyscale
>> to the next two electric field states (the new ones that would appear in the spin-3/2 reoresentation)
>> is the largest local energyscale of the system, and THUS can be ignored. This clearly imposes
>> conditions on the elecrtic energy coupling. The authors write g^2/2 but according to Kogut-Susskind
>> (Ref [55] Equations (6.6) and (7.1)) it can be written as g^2/2a including the lattice spacing a.
>> At the same time, the magnetic energy coupling (the authors simply call it J) is RELATED to this
>> coupling because it should read 4a/g^2. Therefore, for the spin-½ representation to be meaningful
>> in the sense of QED, some constraints on g^2 and on the lattice spacing ‘a’ must be included. The
>> authors should definitely state them.

We thank the Referee for bringing this point up, which is somewhat subtle and not always fully
appreciated. Firstly, we would like to emphasize that the quantum link models generalize the
Wilson-Kogut-Susskind formulation by introducing a gauge invariant parameter, which is the
representation of the quantum links. For Abelian gauge theories, this is the spin-representation
S, such that each value of S defines a completely independent theory. The spin representation
S=1/2 and S=1, for example, defines two independent microscopic theories. If one lives in a "universe"
with the spin-1/2 model, then the electric flux can only take the values E = +1/2 and E = -1/2,
since a spin-1/2 gauge link cannot be raised higher. On the other hand, for the S=1 model, g^2
coupling would have contributed non-trivially since there would have been three states of the
electric flux E = -1, 0, +1. These can also be viewed as independent theories parameterized by S,
where the choice of the representation decides the allowed values. Of course, if one is interested
in achieving the traditional continuum limit, then one should rescale the couplings appropriately,
as is done in arXiv:2104.00025, a related work with some present authors.

Another way of imagining the situation is to recognize that with our microscopic model, we
are sitting on the RG trajectory with g^2 = 0, and can only explore the physics of different
fixed points along this axis. This is a natural way of fine-tuning the problem such that one
cannot move away from this axis.

Regarding the second comment of the Referee, we would like to point out that the electric and the
magnetic field couplings are only related if one flows to a relativistic fixed point. While
this is true for the Wilson-Kogut-Susskind theory in taking the 'usual' continuum limit, it
need not always be the case. In particular, if the system flows to a non-relativistic fixed point,
the magnetic and the electric couplings would typically not be related to each other as in the
relativistic case. Therefore, one should not demand the relation of the couplings beforehand,
a point not always appreciated in the literature. The inclusion of the lattice spacing is therefore
only useful if one flows to the usual relativistic fixed point, where the electric and the magnetic
field scale with the related anomalous dimensions. Since we do not assume that we flow to such a
fixed point, we have deliberately not included the lattice spacing in the couplings.

To summarize, in this work, we aim to explore the physics away from the traditional continuum limit
and therefore do not impose any such restrictions beforehand, or demand the restoration of Lorentz
invariance. In particular, this is done keeping in mind non-relativistic fixed points where
one can flow to. We explain our reasoning in the beginning of Section II.

>> (IV) The Shift symmetry, as it is written in equation (7) does not seem to be correct to me.
>> The bare mass term is not invariant under either of those transformations. Additionally,
>> Susskind’s lattice fermi field is meant to have four sublattices (arranged like a square),
>> each corresponding to a member of the Dirac 4-spinor. Therefore a shift symmetry of one site
>> SHOULD NOT exist (while 2-site in x and/or 2-site in y should). The authors must elaborate on this point.

The shift symmetry is the analog of the continuous chiral symmetry of a Dirac fermion for the
single component staggered fermion. Just as a mass term explicitly breaks the chiral symmetry
of a Dirac fermion, so does the mass term break the shift symmetry of the staggered fermion.
As the Referee correctly points out, a four-component Dirac fermion can be formed using four
single component staggered fermions. In fact, one can show that the single component staggered
fermions (also called Kogut-Susskind fermions in lattice gauge theory community) preserves a
subgroup of the full chiral symmetry of the full Dirac fermions, and is an internal symmetry
in this sense. The two-site lattice translations (in either x or in y direction) is the usual
lattice translation symmetry as also explained in the Manuscript, below the equations 7a,b,c.
We have added a few sentences explaining these issues before equation 7, where the shift symmetry
is introduced.

>> (V) I can not find any mention of the global excess charge, either as a symmetry (which exists)
>> or as a quantum number to be taken care of. The way I see it, if the simulations were finite-length
>> cylinders (that could be done with finite-size DMRG), the authors could control `some’ degree of
>> excess charge, either positive or negative, by setting appropriate conditions on the EM quantum
>> fields at the two edges of the cylinder (aka Von Neumann boundary conditions). However, the authors
>> are tackling the problem for an infinite cylinder (via infinite-DMRG), which, if I understand correctly,
>> automatically leads them to the sector of zero excess charge, the only sector where the gauge field
>> can be really translationally invariant. Is this true? If it is, I believe the authors should include
>> the latter observation in the manuscript.

As the Referee correctly states, we work in the sector of zero excess charge. This is a common
choice in particle physics scenarios, and is explicitly stated in the parageaph below Eq.(6).

>> (VI) Figure 2 is a nice phase diagram, but it is plotted as a function of the effective couplings (T,M,J) .
>> IF the authors care about visibility to the high-energy community, they may consider exhibiting the same
>> plot a second time, but with the bare couplings of the lattice QED theory (QED dimensionless coupling g,
>> lattice spacing a, bare mass m). Conversion rates are: T => c \hbar /( 2a ) ; M => m c^2 ; J = 4 c \hbar /( a g^2 ).
>> You are welcome.

We thank the Referee very much for this suggestion, but---considering the discussion above---we consider
it might be somewhat misleading to plot it
in terms of effective couplings. Firstly, t is the hopping strength and not the temperature, and moreover
the conversions assume the usual relativistic relation between J and g^2, which need not be true. Such a
depiction, in our opinion, would only make sense after the fixed points (especially a second order phase
transition) has been identified, and once the scaling dimensions have been determined appropriately (to
understand especially if it is a relativistic or a non-relativistic fixed point), one can relate the
couplings as proposed. Thus, we prefer to retain only the version with the effective couplings (t,M,J).
However, we do mention the relativistic scaling in the text and also our motivation of not using the
conversion factors, so that it is clear to the reader.

>> (VII) The chiral condensate: Equation (15) would be much easier to read if it would be explained that
>> the whole expression within the sum is simply the modulus of the electrical charge at site j.

We thank the Referee for pointing this out, this has been explained now.

>> (VIII) The flippability operators O_j and Q_j seem to be very interesting, and capable of detecting interesting
>> thermodynamical phases. However, from equation (17) it seems that you are only interested in the Brillouin zone
>> points (0,0) and (\pi,\pi). Since the authors have calculated the whole 2-point correlations functions in real
>> space, they should consider exhibiting the complete static structure factor S(k_x, k_y) \sim
>> sum_{j, j’} < O_j O_(j+j’) > e^{i k \cdot j’}. This way, they should be able to immediately identify if another
>> wavevector is relevant (beside the Brillouin center and corner). Was this check made?
>> ESPECIALLY figure 6 (left panel) seems to point towards an order at the (\pi, 0) point.

The Referee raises a very good point, and we now show the results for the structure factor computed from both
the $O_j$ (eq. 16a) and the $U_\square$ (eq. 5) operators. This is particularly interesting to show for the
proposed spin-liquid phase, which then displays the only ordering at $(0,0)$, showing that no other symmetry
is broken. We have a special appendix E, in which these results are shown in detail, together with the
order parameters in the proposed spin liquid phase. This also answers a query of the second referee. This Appendix
is also refereed to in the main text, to indicate where to look for the details without breaking the flow.

Moreover, Fig 6 (left panel) it should be noted that the links, represented by the arrows, point to the top and
bottom (for vertical links) and left and right (for the horizontal links) as we move one unit in the y-direction
indicating that the order is more (\pi,\pi) instead of (0,0). Similarly the fermion occupation also alternate
between 0 and 1 indicating to the same order. Noting the shift operator introduced earlier, it is easy to
conclude that the discrete chiral symmetry is broken. This is also visible in the structure factors.

>> (IX) Figure 6. I am understanding that Ly = 4 and PBC on y-direction, correct? So, in this case are the top
>> plaquettes and the bottom plaquettes, in figure 6, actually the same plaquettes? If yes, the picture is somewhat
>> unclear. Authors should consider re-editing.

We thank the Referee for pointing out the problem with the earlier figure, which we had missed. We have
now remade this figure and the co-ordinates of the various points are written out to explain the periodicity
in the y-direction.

>> (X) Finally, it could be helpful to have some cartoon picture to intuitively explain, with the help of visual
>> drawing, the prominent feature of the various phases detected by the authors. Figures 8 and 9 attempt to do so,
>> but being actual data plots, they convey the message poorly. Authors should consider sketching an actual drawing.

Thanks to the Referee for pointing this out. We include some cartoon sketches to elaborate these points.

We hope that the Referee is satisfied with our responses to their comments, and would consider recommending
publication of our paper in SciPost Physics.

---

## Round 1 · Referee Report · Anonymous · 2022-4-22

Report

The research shown in the manuscript "Ground-state phase diagram of quantum link electrodynamics in (2+1)-d" is timely and will deserve to be published but after some minor concerns have been addressed.
The manuscript studies the phase diagram of a U(1) lattice gauge model with dynamical matter in a particular formulation given by quantum link models (QLM).
It is particularly interesting because the model interpolates between two limits of U(1) QLM with different Gauss laws.
Several order parameters are used to characterize the phase diagram of the model.
For sure, there is one clear conclusion: there are at least two phases describing the two limiting cases of the model, i.e. the two Gauss laws.
The intermediate region seems to be a more difficult region to analyse numerically with the variational method used in the paper. For instance, the entanglement entropy increases around this region.
It is also in the intermediate region where a third phase is conjectured.
It is about the characterization of this intermediate region where I have the main questions:
- The chiral condensate or the flippable plaquette operator seems to indicate the existence of two phases
- The clockwise plaquette operator (fig. 3c and 5c) do not completely match. I would expect that the lower part of fig. 5c will have some correspondence with fig. 3c, won't they?
- Would it be possible to analyse the model with m=0? or at least some plot from the data that the authors already have?
After these minor questions about the existence and the nature of the third hypothetical phase have been resolved, we will recommend the publication of the manuscript.

  • validity: -
  • significance: -
  • originality: -
  • clarity: -
  • formatting: -
  • grammar: -

Author:  Tomohiro Hashizume  on 2022-05-28  [id 2534]

(in reply to Report 2 on 2022-04-22)

>> The research shown in the manuscript "Ground-state phase diagram of quantum link electrodynamics in (2+1)-d"
>> is timely and will deserve to be published but after some minor concerns have been addressed. The manuscript studies
>> the phase diagram of a U(1) lattice gauge model with dynamical matter in a particular formulation given by quantum
>> link models (QLM). It is particularly interesting because the model interpolates between two limits of U(1) QLM
>> with different Gauss laws.
>> Several order parameters are used to characterize the phase diagram of the model.
>> For sure, there is one clear conclusion: there are at least two phases describing the two limiting cases of the model,
>> i.e. the two Gauss laws. The intermediate region seems to be a more difficult region to analyse numerically with the
>> variational method used in the paper. For instance, the entanglement entropy increases around this region.
>> It is also in the intermediate region where a third phase is conjectured.

We thank the Referee for not only recognizing the importance of the work, but also appreciating the key points of the presented work very clearly. We attempt to answer the points raised by the Referee below.

>> It is about the characterization of this intermediate region where I have the main questions:
>> - The chiral condensate or the flippable plaquette operator seems to indicate the existence of two phases

We completely agree with the Referee for this point. Results for this have been shown in Fig 3a and Fig 5a (top left) for the chiral condensate.

>> - The clockwise plaquette operator (fig. 3c and 5c) do not completely match. I would expect that the lower part
>> of fig. 5c will have some correspondence with fig. 3c, won't they?

Indeed, the Referee is right in pointing this out. There is mislabeled in the Fig 5a, which seems to imply the y-axis in the figures reaches down to J/t=0. In fact, this is not true, and the y-axis reaches down to J/t=0.5 which has now been clearly indicated.

We point to the discussion in the beginning of Section IV, and alluded to throughout Section IV of the qualitative difference in the cases of J/t=0 and J/t > 0. This is similar to an 'order-by-disorder' phenomenon, since at J/t=0 the fluctuations of the gauge field, controlled by the plaquette operator (which is relevant) are absent, but gets turned on even for an infinitesimally small $J/t$ qualitatively changing the phase diagram.

>>- Would it be possible to analyse the model with m=0? or at least some plot from the data that the authors already have?

Thanks to the Referee for motivating us to explain the physics of the model at m/t = 0. We have now added an entire appendix, Appendix E, where we discuss the physics in the regime m/t=0, by showing the order parameters, as well as structure factors (as motivated by the first referee). These results point to the conclusion that it seems to be a strongly interacting liquid phase which does not break any lattice (or internal) symmetries. The Appendix is also referred to in the main text to suggest to the reader to look for the details, but without breaking the flow
of the results.

>> After these minor questions about the existence and the nature of the third hypothetical phase have been resolved,
>> we will recommend the publication of the manuscript.

We hope that our responses and additional provided material convinces the Referee to agree to recommend the manuscript.

Author:  Tomohiro Hashizume  on 2022-05-20  [id 2499]

(in reply to Report 2 on 2022-04-22)

>> The research shown in the manuscript "Ground-state phase diagram of quantum link electrodynamics in (2+1)-d"
>> is timely and will deserve to be published but after some minor concerns have been addressed. The manuscript studies
>> the phase diagram of a U(1) lattice gauge model with dynamical matter in a particular formulation given by quantum
>> link models (QLM). It is particularly interesting because the model interpolates between two limits of U(1) QLM
>> with different Gauss laws.
>> Several order parameters are used to characterize the phase diagram of the model.
>> For sure, there is one clear conclusion: there are at least two phases describing the two limiting cases of the model,
>> i.e. the two Gauss laws. The intermediate region seems to be a more difficult region to analyse numerically with the
>> variational method used in the paper. For instance, the entanglement entropy increases around this region.
>> It is also in the intermediate region where a third phase is conjectured.

We thank the Referee for not only recognizing the importance of the work, but also appreciating the key points
of the presented work very clearly. We attempt to answer the points raised by the Referee below.

>> It is about the characterization of this intermediate region where I have the main questions:
>> - The chiral condensate or the flippable plaquette operator seems to indicate the existence of two phases

We completely agree with the Referee for this point. Results for this have been shown in Fig 3a and Fig 5a
(top left) for the chiral condensate.

>> - The clockwise plaquette operator (fig. 3c and 5c) do not completely match. I would expect that the lower part
>> of fig. 5c will have some correspondence with fig. 3c, won't they?

Indeed, the Referee is right in pointing this out. There is mislabeled in the Fig 5a, which seems to imply the
y-axis in the figures reaches down to J/t=0. In fact, this is not true, and the y-axis reaches down to J/t=0.5
which has now been clearly indicated.

We point to the discussion in the beginning of Section IV, and alluded to throughout Section IV of the qualitative
difference in the cases of J/t=0 and J/t > 0. This is similar to an 'order-by-disorder' phenomenon, since at J/t=0
the fluctuations of the gauge field, controlled by the plaquette operator (which is relevant) are absent, but gets
turned on even for an infinitesimally small $J/t$ qualitatively changing the phase diagram.

>>- Would it be possible to analyse the model with m=0? or at least some plot from the data that the authors already have?

Thanks to the Referee for motivating us to explain the physics of the model at m/t = 0. We have now added an entire
appendix, Appendix E, where we discuss the physics in the regime m/t=0, by showing the order parameters, as well as
structure factors (as motivated by the first referee). These results point to the conclusion that it seems to be
a strongly interacting liquid phase which does not break any lattice (or internal) symmetries. The Appendix is
also referred to in the main text to suggest to the reader to look for the details, but without breaking the flow
of the results.

>> After these minor questions about the existence and the nature of the third hypothetical phase have been resolved,
>> we will recommend the publication of the manuscript.

We hope that our responses and additional provided material convinces the Referee to agree to recommend the manuscript.

---

## Round 2 · Referee Report · Anonymous (Referee 2) · 2022-5-24

Report

After reading the latest version of the manuscript and the changes made by the authors, we recommend the publication of the article.

---

## Round 2 · Referee Report · Anonymous (Referee 1) · 2022-6-2

Strengths

[see the report]

Weaknesses

[see the report]

Report

I have reviewed the new version of the manuscript and the reply from the authors to my comments. Since the authors addressed all my questions on-point and made appropriate changes to the document, I can give my positive recommendation of the manuscript for publication on Scipost.

---

## Round 2 · Author Response

Dear editors,
We would like to thank you for reconsidering our manuscript "Ground-state phase diagram of quantum link electrodynamics in $(2+1)$-d." We would also like to use this opportunity to thank reviewers for the helpful comments for further improving our manuscript.

Revisions are made to the figures and the text to address the reviewer comments. We believe that the revision has led to an improved clarity of the manuscript.

We very much hope that the revised manuscript is accepted for publication in SciPost.

Best regards,

on behalf of the authors, Tomohiro Hashizume

---

## Round 2 · List of Changes

List of revised or added figures : 1. Fig. 2. : Sketch of the phases are added for the quantum dimer and quantum link phase. 2. Fig. 3 a. : Removed an erroneously placed label "J/t". 3. Fig. 5, Fig. 11, and Fig. 12. : Tick label "0.5" is added on the y-axis on all the subfigures to clarify that the smallest value of J/t is 0.5 NOT J/t=0. 4. Fig. 4, Fig. 6, and Fig. 12 c. : Labels "PBC", labels "OBC", and coordinates are added to clarify the imposed boundary conditions and the coordinates of the lattice. 5. Fig. 4, Fig. 6, and Fig. 12 c. : The last sentence in the caption is added for explaining the changes made in the point 4. 6. Fig. 13 : This figure is added with an accompanied caption as a support to Appendix E.

List of revisions in the text : 1. Page 1 , Column 1, Index : Index element "Appendix E: E. Order Parameters and Structure factors at vanishing bare mass" is added (automatically generated). 2. Page 3 , Column 1, Line 5-31 : A paragraph is added. 3. Page 4 , Column 2, Line 9-15 : Two sentences starting from "We note that..." are added. 4. Page 5 , Column 1, Line 17-26 : Three sentences starting from "The symmetry is the..." are added. 5. Page 7 , Column 2, Line 40-42 : One sentence starting from "The expression within..." is added. 6. Page 11, Column 1, Line 6-12 : One sentence starting from "We substantiate this claim..." is added. 7. Page 15, Column 1, Line 7-Page 16, column 2, Line 7 (end) : The entirety of Appendix E "Order Parameters and Structure factors at vanishing bare mass" is added.

---

## Editorial Decision

published